# Ideas and perspectives: How coupled is the vegetation to the boundary layer?

Martin G. De Kauwe[1,2], Belinda E. Medlyn[3], Jürgen Knauer[4], and Christopher A. Williams[5]

[1]ARC Centre of Excellence for Climate Extremes, University of New South Wales, Sydney, NSW 2052, Australia
[2]Department of Biological Science, Macquarie University, North Ryde NSW 2109 Australia
[3]Hawkesbury Institute for the Environment, Western Sydney University, Locked Bag 1797, Penrith NSW 2751 Australia
[4]Department of Biogeochemical Integration, Max Planck Institute for Biogeochemistry, 07745 Jena, Germany
[5]Graduate School of Geography, Clark University, 950 Main Street, Worcester, MA01602 USA

*Correspondence to:* Martin De Kauwe (mdekauwe@gmail.com)

**Abstract.** Understanding the sensitivity of transpiration to stomatal conductance is critical to simulating the water cycle. This sensitivity is a function of the degree of coupling between the vegetation and the atmosphere, and is commonly expressed by the decoupling factor. The degree of coupling assumed by models varies considerably and has previously been shown to be a major cause for model disagreement when simulating changes in transpiration in response to elevated $CO_2$. The degree of coupling also offers us insight into how different vegetation types control transpiration fluxes, fundamental to our understanding of land–atmosphere interactions. To explore this issue, we combined an extensive literature summary from 41 studies, with estimates of the decoupling coefficient estimated from FLUXNET data. We found some notable departures from values previously reported in single site studies. There was large variability in estimated decoupling coefficients (range = 0.05–0.51) for evergreen needleleaf forests. A result that was broadly supported by our literature review, but contrasts with the early literature which suggests evergreen needleleaf forests are generally well-coupled. Estimates from FLUXNET indicated that evergreen broadleaved forests were the most tightly coupled, differing from our literature review, which instead suggested it was evergreen needleleaf forests. We also found that the assumption that grasses would be strongly decoupled (due to vegetation stature) was only true for high precipitation sites. These results were robust to assumptions about aerodynamic conductance and to a lesser extent, energy balance closure. Thus, these data form a benchmarking metric against which to test model assumptions about coupling. Our results identify a clear need to improve the quantification of the processes involved in scaling from the leaf to the whole ecosystem. Progress could be made with targeted measurement campaigns at flux sites, as well as greater site characteristic information across the FLUXNET network.

## 1 Introduction

Predicting the response of transpiration to global change and the subsequent feedback to climate remains a major challenge for Earth system models (Zhu et al., 2017). Improving our understanding of how stomatal controls on transpiration vary between vegetation types is fundamental to simulating land–atmosphere interactions. Experimental evidence strongly indicates that stomatal conductance ($G_s$) is generally reduced in response to elevated $CO_2$ (Morison, 1985; Medlyn et al., 2001; Ainsworth

and Rogers, 2007), due to either a decrease in stomatal aperture with the reduced photosynthetic demand for $CO_2$ and/or a change in stomatal density (McElwain and Chaloner, 1995; Woodward and Kelly, 1995). In models, incorporating this leaf-level reduction in $G_s$ commonly results in predictions of decreased transpiration and increased runoff at global scales (Gedney et al., 2006; Betts et al., 2007; Cao et al., 2010). However, the magnitude of this effect varies strongly among models, because the sensitivity of transpiration to a change in $G_s$ depends on the assumption made about the strength of coupling of the vegetation to the surrounding boundary layer (McNaughton and Jarvis, 1983; Jarvis and McNaughton, 1986; McNaughton and Jarvis, 1991; Jacobs and De Bruin, 1992). De Kauwe et al. (2013) identified differences in the degree of coupling to be a major cause of disagreement among 11 model predictions of transpiration in response to elevated $CO_2$ at two forest Free-Air $CO_2$ Enrichment (FACE) experiments in the USA. Consequently, resolving this discrepancy among models in their assumptions of vegetation-atmosphere coupling is key to reducing model uncertainty in future predictions of changes in transpiration.

The degree of coupling between vegetation and the atmosphere is commonly expressed by the decoupling factor ($\Omega$; Jarvis and McNaughton, 1986). If the decoupling factor is high, transpiration is more strongly controlled by incoming radiation and less by changes in $G_s$. Low stature-canopies, and species with large leaves, are expected to be more decoupled (high $\Omega$), than tall-stature canopies, and species with small leaves. This occurs, because: (i) small-stature canopies decrease the surface roughness, and hence the aerodynamic conductance; and (ii) large leaves decrease the leaf boundary layer conductance. Both act to diminish the rate of exchange between the vegetation surface and the atmosphere. Other characteristics of the vegetation, including, foliage clumping, leaf shape, canopy density and the vertical canopy distribution, will also alter the coupling. Values given in the literature for coniferous forests are typically low, $\Omega = $ ~0.1-0.2 (Whitehead et al., 1984; Jarvis, 1985; Lee and Black, 1993; Meinzer et al., 1993). Values are typically higher for deciduous broadleaved: $\Omega = 0.2$-0.4 (Magnani et al., 1998; Wullschleger et al., 2000), evergreen broadleaved species: $\Omega = 0.4$-0.9 (Meinzer et al., 1997; Wullschleger et al., 1998), grasses: $\Omega = 0.8$ (McNaughton and Jarvis, 1983), and crops: $\Omega = 0.2$-0.9 (Black et al., 1970; Brown, 1976; Meinzer et al., 1993; Mielke et al., 1999). These literature estimates of the degree of coupling are wide and thus, do not offer a clear constraint to models. Furthermore, methods to estimate $\Omega$ often differ across studies, which complicates interpretations about variation across plant functional types. Single studies, that have employed a consistent method to estimate $\Omega$ across multiple species are rare (e.g. Stoy et al., 2006; Khatun et al., 2011).

There has been considerable recent effort to develop better global datasets of stomatal behaviour for use by the modelling community (Lin et al., 2015; Miner et al., 2017). However, constraining the coupling between stomatal conductance and transpiration is equally important. For example, De Kauwe et al. (2015) demonstrated modest changes in transpiration when using the Lin et al. (2015) dataset to constrain the parameterisation of $G_s$ in the Community Atmosphere Biosphere Land Exchange (CABLE) land surface model. The CABLE model assumes a relatively weak level of coupling (De Kauwe et al., 2013). It is likely that models that assume stronger coupling (e.g. the Joint UK Land Environment Simulator, JULES; Best et al., 2011) would obtain different results.

To shed new light on this important question of vegetation-atmosphere coupling, we used eddy-covariance data from FLUXNET to estimate the $\Omega$ coefficient for different plant functional types (PFTs). We aimed to: (i) examine if decoupling

coefficients estimated from FLUXNET were consistent with literature values; and (ii) develop a benchmark metric against which to test model assumptions about coupling.

## 2 Materials and Methods

### 2.1 Flux Data

Half-hourly eddy covariance measurements of the exchange of carbon dioxide, energy and water vapour were obtained from the FLUXNET "La Thuile" Free and Fair dataset (http://www.fluxdata.org). We estimated the degree of decoupling (Jarvis and McNaughton, 1986) as:

$$\Omega = \frac{1 + \epsilon}{1 + \epsilon + \frac{G_a}{G_s}} \tag{1}$$

where $\epsilon = s / \gamma$, s is the slope of the saturation vapour pressure curve at air temperature (Pa K$^{-1}$), $\gamma$ is the psychrometric
constant (Pa K$^{-1}$) and $G_a$ (mol m$^{-2}$ s$^{-1}$) is the aerodynamic conductance.

We estimated values of $G_s$ by inverting the Penman-Monteith equation using measured latent heat (LE) flux for all datasets where the net radiation ($R_n$; W m$^{-2}$) and the frictional velocity ($u_*$; m s$^{-1}$) were available:

$$G_s = \frac{G_a \gamma \lambda E}{s(R_n - G) - (s + \gamma)\lambda E + G_a M_a c_p D} \tag{2}$$

where E (mol m$^{-2}$ s$^{-1}$) is the canopy transpiration, $\lambda$ is the latent heat of vaporisation (J mol$^{-1}$), D (Pa) is the vapour pressure
deficit, G (W m$^{-2}$) is the soil heat flux, $M_a$ (kg mol$^{-1}$) is molar mass of air, $c_p$ is the heat capacity of air (J kg$^{-1}$ K$^{-1}$). At sites where values of G were not available, G was set to zero.

$G_a$ was calculated following Thom (1975):

$$G_a = \frac{c}{\frac{u}{u_*^2} + 6.2u_*^{-\frac{2}{3}}} \tag{3}$$

where the first term in the denominator of Eq. 3 represents the turbulent aerodynamic resistance ($G_{am}$), and the second term
the canopy boundary layer component ($G_b$), c = P / ($R_{gas}$ $T_k$) is a conversion factor from units of m s$^{-1}$ to mol m$^{-2}$ s$^{-1}$, P is atmospheric pressure (Pa), $R_{gas}$ is the gas constant (J mol$^{-1}$ K$^{-1}$), $T_k$ is the air temperature in Kelvin, and u (m s$^{-1}$) is the wind speed.

In our analysis we derived the average (three most productive months) decoupling coefficient, as the focus of our manuscript was on the spatial variability in coupling across FLUXNET. This is likely to be a metric that can most readily be exploited to
assess existing coupling assumptions in models. Future analysis may wish to explore the temporal variability in this metric.

The approach we have taken (similar to Jarvis and McNaughton, 1986) ignores differences between canopy and air temperature (radiative coupling) within the canopy (see Martin, 1989). However, correcting for the longwave radiative conductance ($G_r$) most impacts vegetation with the weakest control on transpiration and as a result, this assumption has little impact on the degree of coupling range for forest species, but may be a factor for other species.

Flux data were first screened as follows: (i) data flagged as "good" (quality control flag "fqcOK" = 1; Williams et al., 2012); (ii) data from the three most productive months, to account for the different timing of summer in the Northern and Southern hemispheres; (iii) daylight hours between 8 am and 4 pm, to account for periods when the vegetation is photosynthesising; (iv) half-hours with precipitation, and the subsequent 48 half-hours, were excluded to minimise the influence of soil evaporation (Law et al., 2002; Groenendijk et al., 2011; Dekker et al., 2016); and (v) data with a $u_* < 0.25$ were excluded to avoid conditions of low turbulence (Sánchez et al., 2010). We also excluded sites classified as mixed-forest, permanent wetlands or those where the PFT was unclassified.

Pressure was estimated using the hypsometric equation based on site elevation data. Where site elevation information was missing, values were gap-filled using the 30-arc second (~1 km) global digital elevation model GTOPO30 data from the United States Geological Survey (USGS). After filtering, 164 sites and 592 site-years remained.

We also tested the sensitivity of estimated values to: (i) errors in $G_a$; and (ii) errors due to a lack of energy balance closure. First, we increased/decreased estimated values of $G_a$ by 30% to examine the sensitivity of $G_s$ values inverted from the Penman-Monteith equation. Secondly, following recommendations by Wohlfahrt et al. (2009), we tested the sensitivity of our results to energy balance closure, by correcting fluxes using the Bowen-ratio method (each half-hourly LE and H flux) based on the available energy (Rn–G) on a longer time scale (three most productive months).

We also replicated our analysis using eddy covariance data taken from the FLUXNET2015 dataset (http://fluxnet.fluxdata.org/data/fluxnet2015-dataset). Figure A1 is a replicate of Fig. 1 and shows the patterns we derived are robust across flux releases.

## 2.2 Results

We summarised previously reported estimates of the decoupling coefficient from 41 studies, in Tables 1 and A2. Broadly speaking, estimated decoupling coefficients from FLUXNET (Fig. 1) differed among PFTs in line with literature values (Tables 1 and A2) and in line with expectations related to vegetation roughness and/or stature. Evergreen needleleaf forests (ENF), which have small leaves, were in general tightly coupled (low $\Omega$), while deciduous broadleaved forests, tropical rain forest (large leaves), grasses and crops (small stature), had a lower degree of coupling (higher $\Omega$). However, there were some notable departures from expectations. Estimates derived from FLUXNET indicated that evergreen broadleaf forests were the most coupled PFT (mean $\Omega = 0.21$) as opposed to the literature review, which suggested ENF PFTs were the most coupled (mean $\Omega = 0.19$). The FLUXNET data also indicated unexpectedly wide ranges for $\Omega$ within PFTs. For grasses, $\Omega$ ranged from 0.02–0.8; the number of low values was particular surprising, given the expectation that shorter stature vegetation would be more decoupled.

The wide range in estimated values for ENF sites was also striking; $\Omega$ extended from 0.05 to 0.51. To attempt to better understand this range, we first separated ENF sites into: (a) sites with a low inter-annual coefficient of variation (20%), indicating consistent year-to-year estimates of the degree of coupling; (ii) sites with a coefficient of variation > 20%, indicating sites with year-to-year variability in coupling; and (iii) sites with only two years of data. This separation was intended to rule out sampling issues. Figure 2 shows that the variability in the estimated decoupling coefficient cannot be explained by sampling bias, with significant site-to-site variability, irrespective of the inter-annual variability.

We then probed these results for relationships with site variables, by testing to see if: (i) sites with higher precipitation (in the three most productive months) were more decoupled, where precipitation was assumed to be a proxy for leaf area index (LAI)/productivity; or (ii) windy sites were more coupled. For grasses we found a significant relationship between the degree of coupling and precipitation (Fig. 3). The data suggest that for sites that are likely to be more open grasslands (i.e. sites with a low precipitation) the vegetation is very coupled to the atmosphere, with a high level of stomatal control. This relationship between the degree of coupling and precipitation (r=0.46) explains the high variability in estimated decoupling coefficients for grasses shown in Fig. 1. The prediction that grasses would be weakly coupled due to small vegetation stature only holds true at sites with high 3-month precipitation, which are presumably sites with high LAI. We also found a significant relationship for ENF sites (r=0.40), and deciduous broadleaved forests (r=0.64) suggesting that the degree of coupling declined with canopy density. We also found evidence of a weak negative relationship (r=−0.21) between wind speed and the degree of coupling for forest sites, i.e. windier sites tended to be more coupled (Fig 4). For non-forest PFTs, we did not find a significant relationship between wind speed and coupling.

Finally, we examined sensitivity of our results to potential errors. We tested whether our results were sensitive to different estimates of $G_a$ and whether our estimates of $G_s$ were sensitive to energy imbalance. We found that the broad pattern of our results in Fig. 1 was insensitive to errors in $G_a$. Increasing or decreasing $G_a$ by 30%, led to the median decoupling coefficient decreasing or increasing by roughly 0.05 for evergreen broadleaf forest (EBF) sites for example. However, we did find that our results were sensitive to a correction for the lack of energy balance closure. Figure A2 shows that attempting to correct for a lack of closure leads to sites becoming less coupled, but does not shift the between-PFT differences in the degree of coupling. The largest changes were for C3 crops ($\Omega$ changed from ~0.44 to ~0.6) and deciduous broadleaved forests ($\Omega$ changed from ~0.31 to ~0.41).

## 2.3 Discussion

Correctly characterising the sensitivity of transpiration to $G_s$ is critical for simulating the water cycle, particularly for future projections of the terrestrial biosphere where it is widely expected that $G_s$ will decrease in response to increasing atmospheric $CO_2$. The parameterisation of this crucial link between leaf– and canopy–scale water fluxes has been largely ignored in model studies addressing the impact of elevated $CO_2$ (Betts et al., 2007; Cao et al., 2010; Zhu et al., 2017). Resulting projections of changes in transpiration and associated fluxes (e.g. runoff, precipitation) are likely to be model-specific, with large uncertainty among models (De Kauwe et al., 2013). Model studies rarely provide information about the degree of coupling assumed within the model. The range of assumptions commonly incorporated in models include: (i) coupling is a function of roughness length

(determined by vegetation height), e.g. JULES; (ii) coupling is a function of leaf size, e.g. CLM (the Community Land Model; Oleson et al., 2013); (iii) coupling is affected by within canopy turbulence, e.g. CABLE (Raupach et al., 1997; Kowalczyk et al., 2006); (iv) some combination of all three, e.g. CABLE/CLM; (v) coupling is not sensitive to wind speed (i.e. wind speed is fixed to 5 m s$^{-1}$), e.g. SDGVM (Sheffield Dynamic Global Vegetation Model; Woodward et al., 1995); or (vi) models that

use an alternative to the Penman-Monteith equation, e.g. LPJ (Lund-Potsdam-Jena family of models; Sitch et al., 2003). This family of models use an empirically calibrated hyperbolic function of canopy conductance (Huntingford and Monteith, 1998) and the implied level of coupling depends on how this function is parameterised.

  Understandably, the pioneering work of Jarvis and McNaughton (1986) is widely cited when issues of coupling are discussed in the literature. However, many of the earlier estimates of coupling they summarised were taken from single sites and thus does

not necessarily reflect the diversity of global vegetation. In this study we have summarised 41 literature studies, in combination with estimates of the decoupling coefficient from 164 sites and 592 site-years from FLUXNET. Our literature summary (Tables 1 and A2) highlights the diversity of approaches used to determine $\Omega$. In contrast, we have applied a consistent methodology across all the 164 FLUXNET sites. For forest PFTs, our results point to a weaker level of coupling than is often assumed. Notably, ENF species were found to be less coupled (mean $\Omega = 0.21$; range = 0.05–0.51) across FLUXNET than summarised

in Jarvis and McNaughton (1986) ($\Omega = 0.1$). Our estimate derived from FLUXNET was supported by our wider literature summary (n=13). We found that the often assumed low degree of coupling for grasses is likely to only be true for high precipitation (and presumably high LAI) sites; low precipitation sites were strongly coupled. A further plausible explanation is that these drier sites are limited by available soil moisture, with lower $G_s$ resulting in a high degree of coupling. We could not easily explain the coupling values estimated for evergreen broadleaf forests, which were estimated to be more coupled

than evergreen needleleaf forests; a break from theoretical understanding developed from vegetation roughness and/or stature. Finally, grouping the data by PFTs also highlighted marked within-PFT variation in the degree of coupling.

  As land models move towards more realistic representations of the variability of stomatal conductance (De Kauwe et al., 2015), informed by leaf-level syntheses (Lin et al., 2015; Miner et al., 2017), it is also important that they accurately simulate the coupling between vegetation and the atmosphere. Without this focus, any efforts to improve the realism at the leaf-scale

will not be reflected in improvements in simulated transpiration at the canopy scale.

### 2.3.1 Caveats

One criticism of the approach taken here is that we have assumed a big-leaf approximation to estimate vegetation the degree of coupling (see Raupach and Finnigan, 1988). It is of course likely that variation within a canopy in terms of micro-climate (i.e. vapour pressure deficit, irradiance, temperature), as well as how stomata respond, may invalidate this approach. Use of a

30 big-leaf approximation could be a possible explanation for the surprisingly high level of coupling found in evergreen broadleaf forests, although it would appear unlikely given the weaker level of coupling found for deciduous broadleaved and tropical rainforest species.

  We found high variation in the estimated coupling factor both across sites and within sites. Two assumptions we make with respect to the flux data could explain this variation. Firstly, we excluded data following rainfall (48 hours) (Law et al., 2002;

Groenendijk et al., 2011; Dekker et al., 2016) to minimise the effects of soil evaporation. Clearly, if soil evaporation is still a component of the LE flux after this point it would introduce error to our estimates. This assumption may also vary with PFT. Secondly, flux towers commonly do not close the energy balance (Foken, 2008; Wilson et al., 2002). Our use of the inverted Penman-Monteith equation implies that we are attributing any errors due to energy imbalance to the sensible heat flux (see Knauer et al., 2017). Additionally, where data on the soil heat flux were missing, we assumed there was no storage. Correcting for these issues is not straightforward as it requires determining which flux is the source of the error (see Wohlfahrt et al., 2009, for a detailed discussion). We followed recommendations by Wohlfahrt et al. (2009) and tested the sensitivity of our results to energy balance closure, by correcting using the Bowen-ratio method based on the available energy (Rn–G). Whilst we did find some sensitivity in our results (particularly for C3 crops and deciduous broadleaved forests), it did not change the ordering of coupling factors between PFTs, or explain the unexpected high level of coupling for EBF sites.

Finally, we estimated the canopy aerodynamic conductance ($G_a$) using an empirical equation following Thom (1975). Knauer et al. (2017) tested the impact on different methods of estimating $G_a$ from flux data on estimates of the stomatal slope parameter (the sensitivity of stomatal conductance to assimilation). They found that a more physically-based representation of $G_a$ (Su et al., 2001), led to lower estimate of $G_a$ at two EBF flux sites, and higher estimates of $G_a$ at another EBF and a deciduous broadleaved site. We tested the sensitivity of our results to a change in $G_a$ of the order shown by Knauer et al. (2017) and found the patterns in coupling to be robust across PFTs.

### 2.3.2 Route forward

Estimates of coupling from ecosystem scale flux data are directly relevant for models. We previously speculated (De Kauwe et al., 2013) that discrepancies among models in coupling might be resolved by examining eddy covariance data. The range in coupling factors we have estimated from the FLUXNET data provides an overall constraint on the degree of coupling that should be assumed in models, as well as an indication of the appropriate degree of variability in coupling across PFTs and rainfall regimes. The next steps involve determining what models currently assume about the degree of coupling and then to determine how flux-derived estimates of coupling would change model predictions.

In this study we examined the long-term average coupling factor. It may also be instructive to consider how estimated coupling factors change across the course of a day or within a season. However, it is likely that such an approach may be more sensitive to noise in the fluxes as well as events such as drought.

Our results also identify a clear need to better understand leaf-to-atmosphere coupling. We need to better understand why coupling factors vary within PFTs. There are a number of plausible explanations, such as drought, diversity of vegetation within a flux footprint, data issues, and it is likely that more detailed site-specific insight will be required to move forward. To assist in better understanding patterns, we will need greater detail in terms of ancillary data from FLUXNET sites. We attempted to probe our results with respect to canopy height and LAI, but for many sites this information was not available. Other potentially useful information would include leaf size, stem density and crown length, and whether canopy height is static or increasing. These data would facilitate more sophisticated approaches to be explored, for example, estimates of $G_b$ based on leaf size (Su et al., 2001). A more fundamental process understanding will require targeted $G_s$ measurements throughout

the canopy, alongside corresponding sap flux measurements in forests and chamber measurements in grasslands. Recently, Medlyn et al. (2017) compared estimates of plant water-use efficiency derived from leaf gas exchange data and eddy flux data for eight sites where these measurements were acquired at the same point in time. They found similarities for DBF and TRF PFTs, but differences for EBF and ENF PFTs. The authors were unable to explain these scaling discrepancies. Further targeted measurements campaigns at flux sites could lead to new knowledge, which would advance our understanding of the processes involved in scaling from the leaf to the canopy.

*Code availability.* All code is freely available from: https://github.com/mdekauwe/flux_decoupling

*Data availability.* All Eddy covariance data are available from: http://fluxnet.fluxdata.org/data/la-thuile-dataset/

*Competing interests.* The authors declare no competing financial interests.

*Acknowledgements.* M. G. De Kauwe was supported by an Australian Research Council (ARC) Linkage grant LP140100232 and acknowledges support from the ARC Centre of Excellence for Climate System Science CE110001028. This work used eddy covariance data acquired by the FLUXNET community and in particular by the following networks: AmeriFlux (U.S. Department of Energy, Biological and Environmental Research, Terrestrial Carbon Program (DE–FG02–04ER63917 and DE–FG02– 04ER63911)), AfriFlux, AsiaFlux, CarboAfrica, CarboEuropeIP, CarboItaly, CarboMont, ChinaFlux, Fluxnet–Canada (supported by CFCAS, NSERC, BIOCAP, Environment Canada, and NRCan), GreenGrass, KoFlux, LBA, NECC, OzFlux, TCOS–Siberia, USCCC. We acknowledge the financial support to the eddy covariance data harmonization provided by CarboEuropeIP, FAO–GTOS–TCO, iLEAPS, Max Planck Institute for Biogeochemistry, National Science Foundation, University of Tuscia, Université Laval and Environment Canada and US Department of Energy and the database development and technical support from Berkeley Water Center, Lawrence Berkeley National Laboratory, Microsoft Research eScience, Oak Ridge National Laboratory, University of California, University of Virginia. Finally, we thank the two anonymous reviewers for their constructive criticisms that improved this work.

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

Table 1: Literature summary of decoupling coefficients, see Table A2 for summaries of individual studies. Plant functional types (PFT) are defined as: ENF - evergreen needle leaved forest, EBF - evergreen broadleaved forest, DBF - deciduous broadleaved forest, TRF - tropical rain forest, SAV - savanna, SHB - shrub, GRA - grasses, C3C - $C_3$ crops, C4C - $C_4$ crops.

| PFT | Mean | Standard Deviation | Min | Max | Number of studies |
|-----|------|--------------------|-----|-----|-------------------|
| ENF | 0.19 | 0.1 | 0.06 | 0.43 | 13 |
| EBF | 0.26 | 0.13 | 0.1 | 0.63 | 12 |
| DBF | 0.36 | 0.18 | 0.1 | 0.7 | 11 |
| TRF | 0.57 | 0.28 | 0.25 | 0.9 | 11 |
| SAV | 0.14 | — | — | — | 1 |
| SHB | 0.27 | 0.19 | 0.13 | 0.4 | 2 |
| GRA | 0.42 | 0.23 | 0.21 | 0.8 | 4 |
| C3C | 0.4 | 0.28 | 0.2 | 0.59 | 2 |
| C4C | 0.58 | — | — | — | 1 |

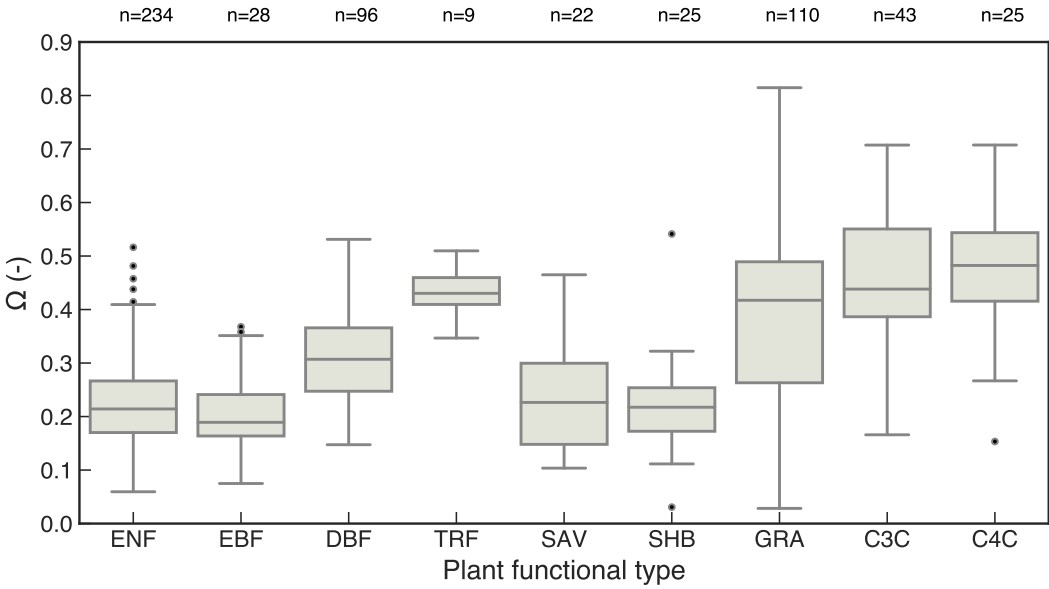

**Figure 1.** Box and whisker plot (line, median; box, inter-quartile range) showing the estimated coupling coefficient ($\Omega$) from FLUXNET data, grouped by plant functional type. Whiskers extend to 1.5 times the inter-quartile range, with dots outside of the whiskers showing outliers. Plant functional types are defined as: ENF - evergreen needle leaved forest, EBF - evergreen broadleaved forest, DBF - deciduous broadleaved forest, TRF - tropical rain forest, SAV - savanna, SHB - shrub, GRA - grasses, C3C - $C_3$ crops, C4C - $C_4$ crops. Values of n indicate the number of site-years for FLUXNET.

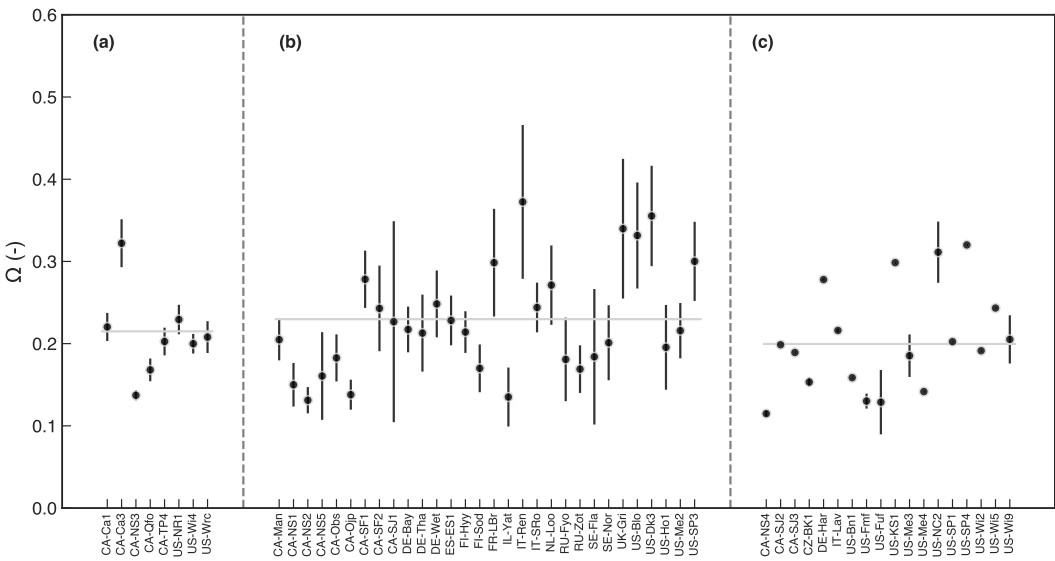

**Figure 2.** Values of the coupling coefficient (Ω) for sites from the evergreen needleleaf forests (ENF) plant functional type. Estimated values of Ω have been split into: (a) sites where the coefficient of variation (COV) is < 20%; (b) sites where the COV is > 20%; and (c) sites with only two years of data. Site errorbars represent one standard deviation (site year variation) in estimated Ω values. Solid horizontal grey lines show overall mean coupling coefficients.

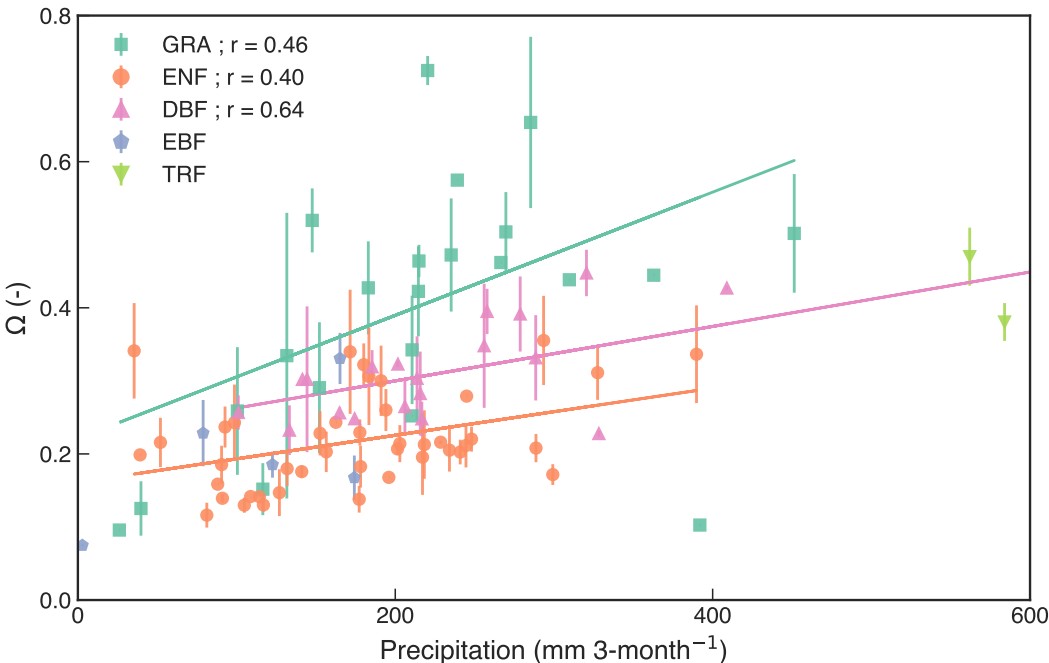

**Figure 3.** Values of the estimated coupling coefficient ($\Omega$) for forest (ENF, EBF, DBF, TRF) vegetation and grasses as a function of precipitation in the three most productive months. Only data were 90% of the three most productive months were flagged as "good" are shown. Lines indicate statistically significant regressions ($P < 0.05$). Plant functional types are defined as: GRA - grasses, ENF - evergreen needle leaved forest, EBF - evergreen broadleaved forest, DBF - deciduous broadleaved forest and TRF - tropical rain forest.

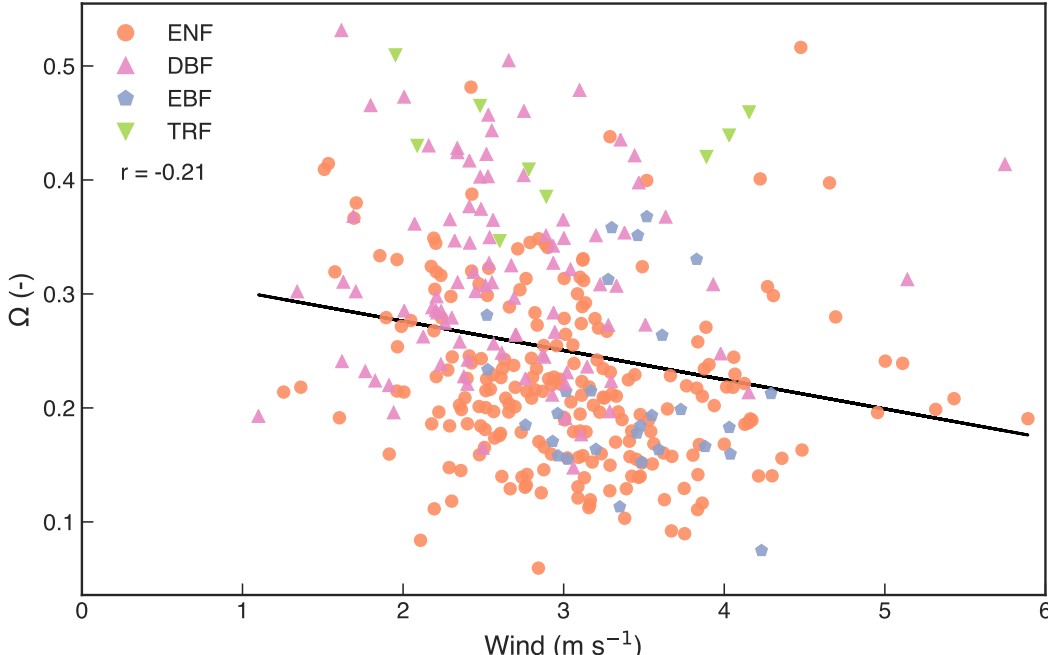

**Figure 4.** Values of the estimated coupling coefficient ($\Omega$) for forest (ENF, EBF, DBF, TRF) vegetation as a function of wind speed. Line indicates statistically significant regression (P < 0.05), r is the correlation coefficient. Plant functional types are defined as: ENF - evergreen needle leaved forest, EBF - evergreen broadleaved forest, DBF - deciduous broadleaved forest, TRF - tropical rain forest.

**Appendix A**

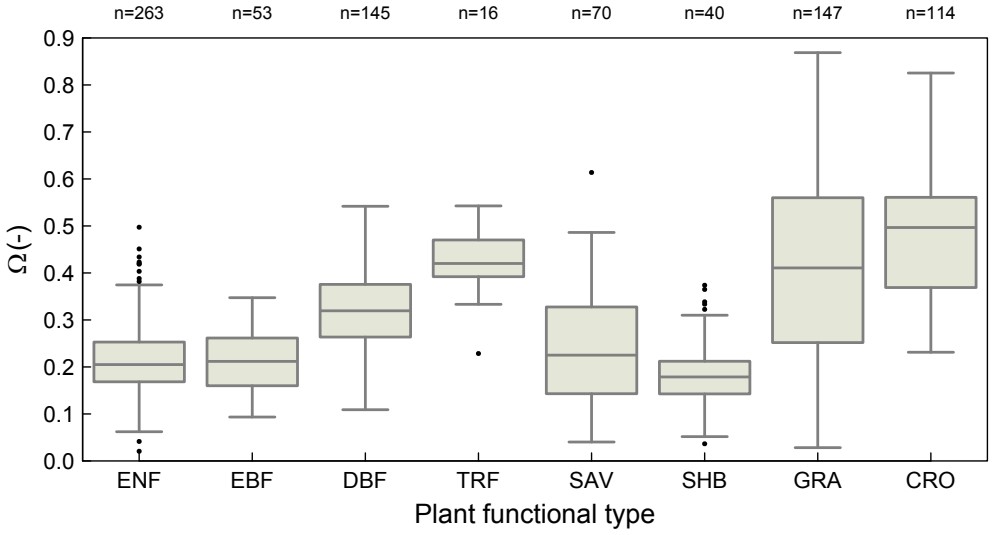

**Figure A1.** Box and whisker plot (line, median; box, inter-quartile range) showing the estimated coupling coefficient ($\Omega$) from FLUXNET2015 data, grouped by plant functional type. Whiskers extend to 1.5 times the inter-quartile range, with dots outside of the whiskers showing outliers. Plant functional types are defined as: ENF - evergreen needle leaved forest, EBF - evergreen broadleaved forest, DBF - deciduous broadleaved forest, TRF - tropical rain forest, SAV - savanna, SHB - shrub, GRA - grasses, C3C - $C_3$ crops, C4C - $C_4$ crops. Values of n indicate the number of site-years for FLUXNET.

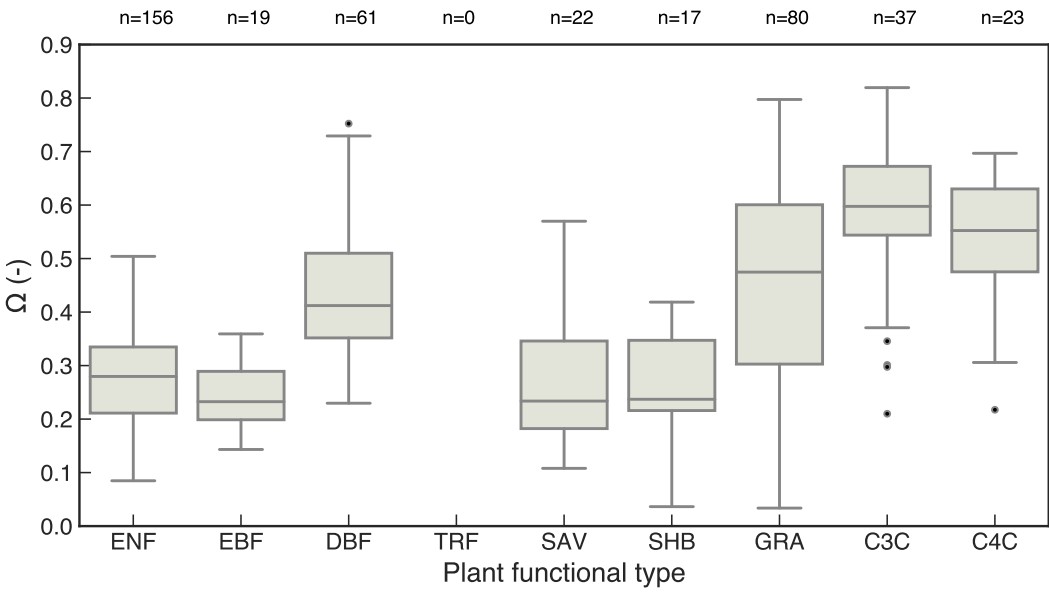

**Figure A2.** Box and whisker plot (line, median; box, inter-quartile range) showing the estimated coupling coefficient ($\Omega$) from FLUXNET data, grouped by plant functional type. These data have been corrected for energy imbalance, adjusting the Bowen-ratio method by the imbalance across the three most productive months. Whiskers extend to 1.5 times the inter-quartile range, with dots outside of the whiskers showing outliers. Plant functional types are defined as: ENF - evergreen needle leaved forest, EBF - evergreen broadleaved forest, DBF - deciduous broadleaved forest, TRF - tropical rain forest, SAV - savanna, SHB - shrub, GRA - grasses, C3C - C$_3$ crops, C4C - C$_4$ crops. Values of n indicate the number of site-years for FLUXNET.

Table A1: FLUXNET site years used in analysis.

| Site ID | Years |
| --- | --- |
| AT-Neu | 2002 2003 2004 2005 |
| AU-How | 2001 2002 2003 2004 2006 |
| AU-Tum | 2001 2002 2003 2004 2005 2006 |
| AU-Wac | 2005 2006 2007 |
| BE-Lon | 2004 2005 2006 |
| BW-Ghg | 2003 |
| BW-Ghm | 2003 |
| BW-Ma1 | 1999 2000 2001 |
| CA-Ca1 | 1998 1999 2000 2001 2002 2003 2004 2005 |
| CA-Ca3 | 2001 2002 2003 2004 2005 |
| CA-Let | 1998 1999 2000 2001 2002 2003 2004 2005 |
| CA-Man | 1994 1995 1997 1998 1999 2000 2001 2002 2003 |
| CA-NS1 | 2002 2003 2004 |
| CA-NS2 | 2002 2003 2004 2005 |
| CA-NS3 | 2001 2002 2004 2005 |
| CA-NS4 | 2003 2004 |
| CA-NS5 | 2001 2002 2003 2004 2005 |
| CA-NS6 | 2001 2002 2003 2004 2005 |
| CA-Oas | 1997 1999 2000 2001 2002 2003 2004 2005 |
| CA-Obs | 1999 2000 2001 2002 2003 2004 2005 |
| CA-Ojp | 1999 2000 2001 2002 2003 2004 2005 |
| CA-Qfo | 2003 2004 2005 2006 |
| CA-SF1 | 2003 2004 2005 |
| CA-SF2 | 2003 2004 2005 |
| CA-SJ1 | 2003 2004 2005 |
| CA-SJ2 | 2005 |
| CA-SJ3 | 2005 |
| CA-TP4 | 2003 2004 2005 |
| CG-Tch | 2006 |
| CH-Oe1 | 2003 2004 2005 |

| Site ID | Years |
| --- | --- |
| CN-Du1 | 2005 2006 |
| CN-Du2 | 2006 |
| CN-HaM | 2002 2003 |
| CN-Xi1 | 2006 |
| CN-Xi2 | 2006 |
| CZ-BK1 | 2004 2005 |
| CZ-BK2 | 2004 |
| DE-Bay | 1996 1997 1998 1999 |
| DE-Geb | 2004 2005 2006 |
| DE-Gri | 2005 2006 |
| DE-Hai | 2000 2001 2002 2003 2004 2005 2006 |
| DE-Har | 2005 2006 |
| DE-Kli | 2004 2005 2006 |
| DE-Meh | 2003 2004 2005 2006 |
| DE-Tha | 1996 1997 1998 1999 2000 2001 2002 2003 2004 2005 |
| DE-Wet | 2002 2003 2004 2005 2006 |
| DK-Fou | 2005 |
| DK-Lva | 2005 |
| DK-Ris | 2004 |
| DK-Sor | 2004 2005 |
| ES-ES1 | 2000 2002 2003 2004 2005 2006 |
| ES-ES2 | 2004 2005 2006 |
| ES-LMa | 2004 2005 2006 |
| ES-VDA | 2004 2005 2006 |
| FI-Hyy | 1998 1999 2000 2001 2002 2003 2004 2006 |
| FI-Sod | 2002 2003 2004 2005 2006 |
| FR-Aur | 2005 |
| FR-Fon | 2005 2006 |
| FR-Gri | 2005 2006 |
| FR-Hes | 1997 1998 1999 2000 2001 2002 2003 2004 2005 2006 |

Table A1 continued from previous page

| Site ID | Years |
|---------|-------|
| FR-LBr | 1996 1997 1998 2000 2003 2004 2005 2006 |
| FR-Lam | 2005 |
| FR-Lq1 | 2005 |
| FR-Lq2 | 2004 2005 |
| FR-Pue | 2000 2001 2002 2003 2004 2005 2006 |
| GF-Guy | 2004 2005 2006 |
| HU-Bug | 2002 2003 2004 2005 2006 |
| HU-Mat | 2004 2005 2006 |
| ID-Pag | 2002 2003 |
| IE-Ca1 | 2004 2005 2006 |
| IE-Dri | 2004 2005 |
| IL-Yat | 2001 2002 2003 2006 |
| IS-Gun | 1996 1997 1998 |
| IT-Amp | 2002 2003 2004 2005 2006 |
| IT-BCi | 2005 2006 |
| IT-Be2 | 2006 |
| IT-Cas | 2006 |
| IT-Col | 1998 1999 2004 2005 2006 |
| IT-Cpz | 1997 2000 2001 2002 2003 2004 2005 2006 |
| IT-LMa | 2004 2005 2006 |
| IT-Lav | 2004 2006 |
| IT-MBo | 2003 2004 2005 2006 |
| IT-Mal | 2004 |
| IT-Noe | 2004 2005 2006 |
| IT-Non | 2001 2002 2003 2006 |
| IT-PT1 | 2003 2004 |
| IT-Ren | 2000 2001 2002 2003 2004 |
| IT-Ro1 | 2000 2002 2003 2004 2005 2006 |
| IT-Ro2 | 2004 2005 2006 |
| IT-SRo | 2000 2002 2003 2004 2005 2006 |

Table A1 continued from previous page

| Site ID | Years |
|---------|-------|
| IT-Vig | 2004 |
| JP-Mas | 2002 2003 |
| NL-Ca1 | 2003 2004 2005 2006 |
| NL-Hor | 2006 |
| NL-Lan | 2005 |
| NL-Loo | 1996 1997 1998 1999 2001 2002 2003 2004 2006 |
| PT-Esp | 2002 2004 2006 |
| PT-Mi1 | 2005 |
| PT-Mi2 | 2004 2005 2006 |
| RU-Cok | 2005 |
| RU-Fyo | 1998 1999 2002 2003 2004 2005 2006 |
| RU-Ha1 | 2003 2004 |
| RU-Zot | 2002 2003 2004 |
| SE-Abi | 2005 |
| SE-Fla | 1996 1997 1998 2001 2002 |
| SE-Nor | 1996 1997 1998 2005 |
| UK-EBu | 2004 |
| UK-ESa | 2004 2005 |
| UK-Gri | 1998 2000 2001 2006 |
| UK-Ham | 2004 2005 |
| UK-Her | 2006 |
| UK-PL3 | 2005 2006 |
| US-ARM | 2003 2004 2005 2006 |
| US-Aud | 2002 2003 2004 2005 2006 |
| US-Bar | 2004 2005 |
| US-Bkg | 2004 2005 2006 |
| US-Blo | 1997 1998 1999 2000 2001 2002 2003 2004 2005 |
| US-Bn1 | 2003 |
| US-Bn2 | 2003 |
| US-Bo1 | 1996 1997 1998 1999 2000 2001 2002 2003 2004 2005 2006 2007 |

Table A1 continued from previous page

| Site ID | Years |
|---------|-------|
| US-Bo2 | 2004 2005 2006 |
| US-CaV | 2004 2005 |
| US-Dk1 | 2001 2002 2003 2004 2005 |
| US-Dk2 | 2003 2004 |
| US-Dk3 | 2001 2002 2003 2004 2005 |
| US-FPe | 2000 2001 2002 2003 2004 2005 2006 |
| US-Fmf | 2005 2006 |
| US-Fuf | 2005 2006 |
| US-Fwf | 2005 2006 |
| US-Goo | 2002 2003 2004 2005 2006 |
| US-Ha1 | 1992 1993 1994 1995 1996 1997 1998 1999 2000 2001 2003 2005 2006 |
| US-Ho1 | 1996 1997 1998 1999 2000 2001 2002 2003 |
| US-IB1 | 2005 2006 2007 |
| US-IB2 | 2004 2006 2007 |
| US-KS1 | 2002 |
| US-KS2 | 2002 2003 2004 2005 2006 |
| US-MMS | 2000 2001 2002 |
| US-MOz | 2004 2005 2006 |
| US-Me2 | 2003 2004 2005 |
| US-Me3 | 2004 2005 |
| US-Me4 | 2000 |
| US-NC2 | 2005 2006 |
| US-NR1 | 1999 2000 2002 2003 |
| US-Ne1 | 2001 2002 2003 2004 2005 |
| US-Ne2 | 2001 2002 2003 2004 2005 |
| US-Ne3 | 2001 2002 2003 2004 |
| US-SO2 | 1997 1998 1999 2004 2005 2006 |
| US-SO3 | 1997 1998 2005 2006 |
| US-SP1 | 2005 |
| US-SP3 | 1999 2000 2001 2002 2003 2004 |

Table A1 continued from previous page

| Site ID | Years |
|---------|-------|
| US-SP4 | 1998 |
| US-SRM | 2004 2005 2006 |
| US-Ton | 2001 2002 2003 2004 2005 2006 |
| US-UMB | 1999 2000 2001 2002 2003 |
| US-Var | 2001 2002 2003 2004 2005 2006 |
| US-WCr | 1999 2000 2001 2002 2003 2004 2005 2006 |
| US-Wi1 | 2003 |
| US-Wi2 | 2003 |
| US-Wi4 | 2003 2004 2005 |
| US-Wi5 | 2004 |
| US-Wi9 | 2004 2005 |
| US-Wkg | 2004 2005 2006 |
| US-Wrc | 1998 1999 2000 2001 2002 2004 2005 2006 |
| VU-Coc | 2001 2002 2003 2004 |

Table A2: Literature summary of decoupling coefficients. Where possible we have summarised data from the growing season during daylight hours. Where E is transpiration, $G_a$ is the total aerodynamic conductance ($G_a = G_{am} + G_b$), $G_{am}$ is the turbulent aerodynamic resistance, $G_b$ is the canopy boundary layer conductance, $G_s$ is the stomatal condutance, $u_*$ is the frictional velocity, EC is eddy covariance, PM is Penman-Monteith, $R_{tot}$ is the total resistance, $R_a$ is the aerodynamic resistance, PAR is Photosynthetically Active Radiation and TC is the total conductance. The simple gradient approach refers to an estimation of $G_s$ from vapour pressure deficit, pressure, and transpiration. Method refers to the approach to estimating $\Omega$: (1) - default, as in this manuscript (amphistomatous vegetation); (2) hypostomatous vegetation; and (3) accounting for radiative coupling following Martin (1989). Plant functional types (PFT) are defined as: ENF - evergreen needle leaved forest, EBF - evergreen broadleaved forest, DBF - deciduous broadleaved forest, TRF - tropical rain forest, SAV - savanna, SHB - shrub, GRA - grasses, C3C - $C_3$ crops, C4C - $C_4$ crops.

| PFT | Dominant species | $\Omega$ | Scale | Method | $G_a$ | $G_b$ | $G_s$ | Reference |
|---|---|---|---|---|---|---|---|---|
| ENF | *Abies amabilis* | 0.18 | Stand | 2 | f(wind, roughness, radiation) | f(wind) | Inverted PM with sap flow | Martin et al. (2001) |
| ENF | *Callitris glaucophylla* J.Thompson | 0.15 | Canopy | 1 | f(wind, roughness) | — | Inverted PM, *E* from sap flow | Zeppel and Eamus (2008) |
| ENF | *Chamaecyparis obtusa* | 0.21 | Canopy | 1 | f(wind, $u_*$) | — | Inverted PM, *E* from EC | Kosugi et al. (2007) |
| ENF | *Picea glauca* | 0.4 | Stand | 1 | f(wind, $u_*$) | — | Simplified PM with sap flow | Bladon et al. (2006) |
| ENF | *Picea abies* | 0.19 | Canopy | 1 | Inversion bulk transfer of sensible heat (EC) | — | Inversion bulk transfer of sensible and latent heat (EC) | Goldberg and Bernhofer (2008) |
| ENF | *Picea crassifolia* | 0.06 | Canopy | 1 | f(wind, $u_*$) | — | Inverted PM, *E* from EC | Gaofeng et al. (2014) |
| ENF | *Pinus elliotti* | 0.43 | Canopy | 1 | f(wind, roughness) | f($u_*$) | Inverted PM, *E* from EC | Bracho et al. (2008) |

Table A2 continued from previous page

| PFT | Dominant species | $\Omega$ | Scale | Method | $G_{am}$ | $G_b$ | $G_s$ | Reference |
|-----|------------------|----------|-------|--------|----------|-------|-------|-----------|
| ENF | *Pinus pinaster* | 0.08 | Stand | 1 | Empirical reln. between $G_a$ & wind | — | Inverted PM, $E$ from sap flow | Loustau et al. (1996) |
| ENF | *Pinus sylvestris* | 0.1 | Stand | 1 | f(wind, height) | — | Leaf gas exchange | Whitehead et al. (1984) |
| ENF | *Pinus sylvestris* | 0.32 | Canopy | 1 | f(wind, $u_*$) | — | Inverted PM, $E$ from EC | Launiainen (2010) |
| ENF | *Pinus taeda* | 0.25 | Canopy | 1 | f(roughness, $u_*$) | f(characteristic leaf dimension, wind) | Bottom-up model: f(VPD, LAI & radiation) | Stoy et al. (2006) |
| ENF | *Pseudotsuga menziesii* | 0.26 | Canopy | 1 | f(wind, $u_*$) | — | Inverted PM, $E$ from EC | Jassal et al. (2009) |
| ENF | *Pseudotsuga menziesii* | 0.15 | Canopy | 1 | f(wind, $u_*$) | — | Inverted PM, $E$ from EC | Lee and Black (1993) |
| EBF | *Acacia ampliceps* | 0.28 | Stand | 3 | Empirical relationship between $G_a$ & wind speed | — | Inverted (simplified) PM, $E$ from sap flow | Mahmood et al. (2001) |
| EBF | *Azadirachta indica* | 0.28 | Tree | 2 | f(leaf temperature) | — | Inverted (Simplified) PM from sap flow | Smith et al. (1998) |
| EBF | *Citrus limon* | 0.12 | Stand | 1 | f(wind, $u_*$) | — | $R_{tot} - R_a$, $R_{tot}$ from simplified PM, E from sap flow | Nicolás et al. (2008) |

Table A2 continued from previous page

| PFT | Dominant species | $\Omega$ | Scale | Method | $G_{am}$ | $G_b$ | $G_s$ | Reference |
|-----|-----|-----|-----|-----|-----|-----|-----|-----|
| EBF | *Eucalyptus camaldulensis* | 0.33 | Stand | 3 | Empirical relationship between $G_a$ & wind speed | — | Inverted (simplified) PM, *E* from sap flow | Mahmood et al. (2001) |
| EBF | *Eucalyptus crebra* F.Muell. | 0.19 | Canopy | 1 | f(wind, roughness) | — | Inverted PM, *E* from sap flow | Zeppel and Eamus (2008) |
| EBF | *Eucalyptus globulus* | 0.63 | Stand | 1 | TC – $G_s$ | — | Simple gradient approach, *E* from sap flow | White et al. (2000) |
| EBF | *Eucalyptus grandis* | 0.28 | Stand | 1 | f(wind, $u_*$) | — | $R_{tot}$ – $R_a$, $R_{tot}$ from simplified PM, E from sap flow | Mielke et al. (1999) |
| EBF | *Eucalyptus urophylla* | 0.1 | Stand | 1 | f(wind, roughness) | — | Inverted PM, *E* from sap flow | Zhang et al. (2016) |
| EBF | *Nothofagus fusca* | 0.24 | Stand | 1 | f(wind, roughness) | — | Stomatal conductance from gas exchange upscaled by leaf area | Köstner et al. (1992) |
| EBF | *Quercus* | 0.3 | Canopy | 1 | f(roughness, $u_*$) | f(characteristic leaf dimension, wind) | Bottom-up model: f(VPD, LAI & PAR) | Stoy et al. (2006) |

| PFT | Dominant species | $\Omega$ | Scale | Method | $G_{am}$ | $G_b$ | $G_s$ | Reference |
|---|---|---|---|---|---|---|---|---|
| EBF | *Schima superba* | 0.22 | Stand | 1 | f(wind, roughness) | — | Inverted PM, *E* from sap flow | Zhang et al. (2016) |
| EBF | *Vaccinium vitis-vidaea* | 0.2 | Canopy (under-storey only) | 1 | f(wind, $u_*$) | — | Inverted PM, *E* from EC | Iida et al. (2009) |
| DBF | *Acer rubrum* | 0.23 | Stand | 1 | f(wind, $u_*$) | f($u_*$) | Inverted PM, *E* from sap flow | Wullschleger et al. (2000) |
| DBF | *Betula papyrifera* | 0.36 | Stand | 1 | f(wind, $u_*$) | — | Simplified PM from sap flow | Bladon et al. (2006) |
| DBF | *Fagus sylvatica* | 0.28 | Canopy | 1 | f(wind, roughness) | f(leaf size) | Bottom-up model: f(VPD, maximum $G_s$, temperature and radiation) | Magnani et al. (1998) |
| DBF | *Fagus crenata* | 0.3 | Stand | 1 | f(wind, roughness) | f(characteristic leaf dimension, wind) | Inverted PM, *E* from sap flow | Tateishi et al. (2010) |
| DBF | *Fagus sylvatica* | 0.2 | Canopy | 1 | f(wind, roughness) | — | Inverted PM, *E* from the Bowen ratio | Herbst (1995) |

| PFT | Dominant species | $\Omega$ | Scale | Method | $G_{am}$ | $G_b$ | $G_s$ | Reference |
|-----|------------------|----------|-------|--------|----------|-------|-------|-----------|
| DBF | *Juglans regia* | 0.37 | Tree | 2 | — | f(leaf temperature, roughness) | Modelled following Jarvis (1976) and upscaled | Daudet et al. (1999) |
| DBF | *Populus balsamifera* | 0.4 | Stand | 1 | f(wind, u$_*$) | — | Simplified PM with sap flow | Bladon et al. (2006) |
| DBF | *Populus trichocarpa x P. deltoides* | 0.66 | Canopy | 3 | TC – G$_s$ | — | Simple gradient approach, *E* from sap flow | Hinckley et al. (1994) |
| DBF | *Quercus petraea* | 0.1 | Canopy | 1 | f(wind, roughness) | — | Inverted PM, *E* from EC | Granier and Bréda (1996) |
| DBF | *Salix viminalis* | 0.7 | Canopy | 1 | Inverted PM when the canopy is wet | — | Inverted PM, *E* from sap flow | Lindroth (1993) |
| DBF | — | 0.41 | Canopy | 1 | Surface layer similarity | f(u$_*$) | Inverted PM, *E* using the Bowen ratio (EC) | Wilson and Baldocchi (2000) |
| TRF | *Anacardium excelsum* | 0.75 | Tree | 2 | TC – G$_s$ | — | Leaf gas exchange | Meinzer et al. (1993) |
| TRF | *Cecropia longipes* | 0.9 | Tree | 2 | TC – G$_s$ & stomatal conductance | — | Leaf gas exchange | Meinzer et al. (1997) |
| TRF | *Ficus insipida* | 0.82 | Tree | 2 | TC — G$_s$ | — | Leaf gas exchange | Meinzer et al. (1997) |
| TRF | *Hedyosmum anisodorum* Todzia | 0.37 | Leaf | 1 | — | f(wind, leaf extension) | Leaf gas exchange | Motzer et al. (2005) |

| PFT | Dominant species | $\Omega$ | Scale | Method | $G_{am}$ | $G_b$ | $G_s$ | Reference |
|-----|------------------|----------|-------|--------|----------|-------|-------|-----------|
| TRF | *Luehea seemannii* | 0.88 | Tree | 2 | TC – $G_s$ | — | Leaf gas exchange | Meinzer et al. (1997) |
| TRF | *Naucleopsis* sp. | 0.27 | Leaf | 1 | — | f(wind, leaf extension) | Leaf gas exchange | Motzer et al. (2005) |
| TRF | *Psychotria brachiata* Ruiz & Pav. | 0.27 | Leaf | 1 | — | f(wind, leaf extension) | Leaf gas exchange | Motzer et al. (2005) |
| TRF | *Ruagea cf. pubescens* H. Karst. | 0.25 | Leaf | 1 | — | f(wind, leaf extension) | Leaf gas exchange | Motzer et al. (2005) |
| TRF | *Spondias mombin* | 0.9 | Tree | 2 | TC — $G_s$ | — | Leaf gas exchange | Meinzer et al. (1997) |
| TRF | *Trichilia guianensis* Klotzsch | 0.43 | Leaf | 1 | — | f(wind, leaf extension) | Leaf gas exchange | Motzer et al. (2005) |
| TRF | — | 0.43 | Canopy | 1 | TC – $G_a$ – f(wind, $u_*$) | — | Inverted PM, $E$ from EC | Kumagai et al. (2004) |
| SAV | — | 0.14 | Stand | 1 | f(wind, roughness) | — | Inverted PM, $E$ from EC | San José et al. (1995) |
| SHB | *Quer juliflora* | 0.13 | Stand | 3 | Empirical relationship between $G_a$ & wind speed | — | Inverted (simplified) PM, $E$ from sap flow | Mahmood et al. (2001) |
| SHB | *Quercus sp.* | 0.4 | Canopy | 1 | f(wind, roughness) | f($u_*$) | Inverted PM, $E$ from EC | Bracho et al. (2008) |
| GRA | *Brachiaria brizantha* | 0.5 | Canopy | 1 | f(wind, roughness) | f($u_*$) | Inverted PM, $E$ from EC | Meirelles et al. (2011) |

| PFT | Dominant species | $\Omega$ | Scale | Method | $G_{am}$ | $G_b$ | $G_s$ | Reference |
|---|---|---|---|---|---|---|---|---|
| GRA | *Festuca arundinaria* Shreb. | 0.34 | Canopy | 1 | f(roughness, $u_*$) | f(characteristic leaf dimension, wind) | Bottom-up model: f(VPD, LAI & radiation) | Stoy et al. (2006) |
| GRA | *Phragmites australis* | 0.48 | Canopy | 1 | f(wind, $u_*$) | f($u_*$) | Inverted PM, *E* from EC | Zhou et al. (2010) |
| GRA | — | 0.45 | Canopy | 1 | f(wind, $u_*$) | — | Inversion bulk transfer of sensible and latent heat (EC) | Goldberg and Bernhofer (2008) |
| GRA | — | 0.49 | Canopy | 1 | f(wind, $u_*$) | f($u_*$) | Inverted PM, *E* from EC | Aires et al. (2008) |
| GRA | — | 0.31 | Canopy | 1 | f(wind, $u_*$) | f($u_*$) | Inverted PM, *E* from EC | Hao et al. (2007) |
| GRA | — | 0.21 | Canopy | 1 | f(wind, $u_*$) | f($u_*$) | Inverted PM, *E* from EC | Wever et al. (2002) |
| GRA | — | 0.8 | — | 1 | — | — | I— | McNaughton and Jarvis (1983) |
| C3C | *Crotalaria juncea* | 0.59 | Canopy | 1 | f(wind, roughness) | — | Inverted PM, *E* from Bowen Ratio energy balance method | Takagi et al. (2009) |

Table A2 continued from previous page

| PFT | Dominant species | $\Omega$ | Scale | Method | $G_{am}$ | $G_b$ | $G_s$ | Reference |
|---|---|---|---|---|---|---|---|---|
| C3C | *Musa sp.* | 0.2 | Stand | 1 | f(wind, characteristic leaf dimension, LAI) | — | Inverted PM, *E* from sap flow Ratio energy balance method | Haijun et al. (2015) |
| C4C | *Zea mays* | 0.58 | Canopy | 1 | f(wind, roughness) | $f(u_*)$ | Inverted PM, *E* from Bowen Ratio energy balance method | Steduto and Hsiao (1998) |