# Peer review of "Ideas and perspectives: How coupled is the vegetation to the boundary layer?"

_Biogeosciences, 2017_

## Referee Comment (RC1) · Anonymous Referee #1 · 4 Jun 2017

This paper leverages the new FLUXNET2015 dataset to estimate differences in the decoupling coefficient across plant funcitonal types, with some additional discussion of how the coefficient varies in response to canopy structure and meteorological condition. The work builds off a previous study that highlighted the decoupling coefficient as a significant source of uncertainty in some model predictions (De Kauwe et al. 2013). The authors report that evergreen forests are more decoupled than previously thought, and that the decoupling of grasslands depends on mean annual precipitation (among other results).

Overall, I think this analysis will be of interest to members of the observational and modeling communities, and the article is generally well written and the figures are clearly presented. I do have a few suggestions for the authors that would allow them to

bridge what I perceive to be a bit of a gap between the rational/objective of the paper and the interpretation of results.

First, the authors aimed to "examine if decoupling coefficients from FLUXNET were consistent with the literature values." However, the comparison of the decoupling coefficients derived from FLUXNET data and literature values was largely qualitative. The comparison would be more informative if values reported in the literature (or assumed by the models) were presented alongside those derived from the Flux data (for example, by including a bit more information in the box and whisker plots of Figure 1).

Second, the authors aimed to "develop a benchmark metric against which to test model assumptions about decoupling." Presumably this "benchmark metric" is the range of decoupling coefficients presented in the results. Would it be possible for the authors to demonstrate, at least at a few sites, that using a decoupling coefficient informed by the results of this study indeed improves agreement between the predictions of at least one model, and observations (for example, flux tower observations of ET)?

I was also curious about the author's choice to limit the analysis to relatively windy periods between 800 and 1600 hours. Coupling should be greater during these condition when compared to relatively stable conditions, for example those experienced from late evening to sunrise. Do the models similarly use a decoupling coefficient that is most appropriate for those conditions, or do they perhaps employ a lower value that is representative of daytime and nighttime periods (particularly if the models run at a daily timestep)? Further, I though the authors might have missed an opportunity to leverage the high-frequency data from FLUXNET to say something about temporal variation in decoupling over the course of a typical day.

Finally, in paragraph 10, the authors state that LAI information for many sites is not available. Many FLUXNET sites have high-quality ground-based LAI measurements that are not reported to the network. Sometimes an email to the site PIs can turn up useful ancillary data.

---

## Referee Comment (RC2) · Anonymous Referee #2 · 6 Jul 2017

This manuscript presents results from a FLUXNET based analysis on vegetation-atmosphere coupling of transpiration using the omega factor by Jarvis & McNaughton. Aggregating daytime data during the peak growing season across plant functional types (PFT), it was found that evergreen needleleaf forests (ENF) have a lower degree of coupling, and that evergreen broadleaf forests (EBF) and shrubs were more coupled then previously suggested in the literature. The manuscript concludes that this decoupling analysis based on FLUXNET data can be used for benchmarking to test models. The manuscript is overall well written (particularly the Discussion section) and the presented research is of significant scientific interest to improve model estimates of biosphere-atmosphere exchange. Nonetheless, I do have some concerns regarding the argumentation and analysis presented here and would strongly encour-

age the authors to consider the following points, before a revised manuscript could be recommended for publication.

Main Points:

(1) While the manuscript is overall focused on the coupling of vegetation and atmosphere regarding transpiration, the manuscript incoherently switches between the use of the degree of coupling and decoupling, which all refer to omega values between 0 and 1. Although this is linked to the original work by Jarvis & McNaughton (i.e. the decoupling factor), it seems rather confusing for readers of this manuscript and I would suggest using a consistent terminology throughout the manuscript, e.g. the degree of coupling with high omega values referring to a lower degree of coupling.

(2) As the manuscript heavily relies on turbulence based measurements from FLUXNET, there is a high chance that the coupling terminology might be misunderstood. It would help and strengthen the manuscript to more clearly differentiate in the Introduction section, if your terminology of coupling is referring to turbulence conditions above the plant canopy (e.g. quantified by u* or sigma w) or to plant physiological coupling at the leaf level or within the canopy, or between different layers of the canopy such as in forests and woody shrublands. This seems also important to differentiate between the leaf and ecosystem scale in this manuscript as EC flux measurements are at the ecosystem scale, yet some of the presented concepts here are referring to the stomatal coupling at the leaf scale (typically measured by leaf chamber).

(3) The manuscript currently relies substantially on comparisons of FLUXNET derived values to the literature, yet the literature values are not presented and analysed quantitatively. I would suggest considering a figure or table comparing both by PFT and documenting details of the so heavily referred to values from the literature, e.g. on how these were assessed/derived (single site/plant experiment, multiple sites, chambers, EC, season etc) to give readers a better idea of their origin and meaning. The manuscript draws substantial conclusions from the comparison to the literature values

and these needs to be justified accordingly in a quantitative way that is clearly visualized.

(4) The FLUXNET La Thuile data used here is relatively outdated (from 2007) and only includes a limited number of sites (as Free and Fair use subset). Yet the newer and more extensive FLUXNET2015 dataset is available since late 2015 (same website as referred to in Methods section), but including many more sites and site years compared to the 2007 La Thuile dataset (∼1000 vs. ∼1500 site years)ÂÍ, and also including a subset with a similar data policy (TIER1). I am wondering what the reasoning behind this choice of older dataset was and if the manuscript would not benefit from the larger sampling available in the newer dataset, particularly in terms of important PFTs (e.g. TRF) that were poorly represented in the 2007 dataset? It would also benefit the manuscript to have a table of the eventually retained sites (after data screening – see Section 2.1), their used site years and PFT etc. in the Appendix, something that is typically recommended when using the FLUXNET dataset.

(5) The manuscript correctly states (Section 2.3.1) that soil evaporation would bias the coupling estimates, yet it is assumed that this only matters 24 hours after rainfall. In fact soil evaporation is a substantial component of the measured ET at almost all sites and except in closest canopy forests with high LAI, easily contributes up to 50% of total ET, particularly in grasslands and shrublands. Consequently, the bias of soil evaporation on the results of certain PFTs is likely much higher and this needs to be addressed in the interpretation of the Results.

(6) The analysis on the controls of omega is largely focused on wind and precipitation, yet soil moisture and VPD seem much better and more direct controls of plant water stress affecting stomatal conductance. These data are available for most of the sites in the FLUXNET dataset and I would encourage the authors to consider extending their analysis to these controls, and linking these results to the recent literature on stomatal conductance.

Overall, I am aware of the length limitations of Opinion & Perspectives papers, yet a full length manuscript might be more fitting for this study to sufficiently document the analysis and the Conclusions that could be drawn from it.

Specific Comments:

- Page 1, Line 19: please consider adding short explanation why Gs is reduced with elevated CO2.

- It would help to add some details in Section 2.1. why the flux data were screened this way and how this affects the interpretation of your Results. It would also be helpful to specify that your analysis is presenting mean decoupling values during the peak growing season somewhere in the Results.

- Page 4, Line 29: why are open grasslands necessarily sites with low precipitation?

- Page 4, Line 30: or are grasslands just more couple because of having just 1 canopy layer (compared to typically 2 in forest)?

- Page 5, Line 20: please consider removing "low" for consistency.

- Page 5, Line 21: SDGVM = Sheffield Dynamic Global Vegetation Model (add Global)

- Page 5, Line 30: it seems incorrect to write "all" FLUXNET sites her, as you are (i) only using a subset from the 2007 dataset and (ii) further reduce this subset by data screening (see Section 2.1).

- Page 5, Line 30: I would argue that "forest species" is not the correct term here as you are referring to PFTs, not species groups, and the flux measurements are at the ecosystem scale.

- Page 5, Line 31: consider limiting "..the FLUXNET network.." to "FLUXNET".

- Page 6, Line 26-27: Ref. Knauer et al. missing in Reference list, and similarly the incomplete citation of Knauer et al. in Line 31-32.

- Page 6, Line 32: "that" seems redundant here

- Section 2.3.1: what about the limitations arising from the use of an older dataset (despite availability of newer dataset, which poorly represents some PFTs?

- Page 7, Line 8-9: what about general variability of environmental conditions and water availability?

- Page 7, Line 11: the BADM data of the new FLUXNET dataset is more extensive then previously and includes details on canopy height and LAI for many sites

- Page 7, Line 16-17: please specify how process understanding from leaf to canopy scale can be improved, if all the listed measurements are referring to the individual plant and ecosystem scale. Furthermore, such targeted Gs measurements have been performed at various sites already and it is not clear to me what new aspects the authors are suggesting here.

- Figure 1: C4 PFTs in caption but not displayed in Figure? Please add missing data or specify why these are not displayed. Ditto in Figure A1.

- Figure 2: please consider (i) moving site names outside graph as axis caption (i.e. this is a categorical axis), (ii) separating the three groups a-c by vertical lines, (iii) removing selective ticks on x-axis OR adding one for every single site, and (iv) adding details on the meaning of the whiskers in the caption text.

- Figure 3: please consider changing the colours so that these are easier to differentiate, and to change the symbols (i.e. different symbol for each PFT, and potentially increasing size). It could also help to differentiate each regression line with dashed/dotted display.

- Figure 4: why are the C3 grasses displayed in Fig. 3, yet not here? Also, what about croplands? I would also suggest to consider add the slope values here and in Fig. 3 for the regression lines.

[Figure]

---

## Author Comment (AC2) · 20 Jul 2017

We thank the reviewer for their constructive comments and we address their various concerns below. Referee comments are highlighted (R), with our response below (A) in each case.

R: This manuscript presents results from a FLUXNET based analysis on vegetation-atmosphere coupling of transpiration using the omega factor by Jarvis & McNaughton. Aggregating daytime data during the peak growing season across plant functional types (PFT), it was found that evergreen needleleaf forests (ENF) have a lower degree of coupling, and that evergreen broadleaf forests (EBF) and shrubs were more coupled then previously suggested in the literature. The manuscript concludes that

this decoupling analysis based on FLUXNET data can be used for benchmarking to test models. The manuscript is overall well written (particularly the Discussion section) and the presented research is of significant scientific interest to improve model estimates of biosphere-atmosphere exchange. Nonetheless, I do have some concerns regarding the argumentation and analysis presented here and would strongly encourage the authors to consider the following points, before a revised manuscript could be recommended for publication.

Main Points: R: (1) While the manuscript is overall focused on the coupling of vegetation and atmosphere regarding transpiration, the manuscript incoherently switches between the use of the degree of coupling and decoupling, which all refer to omega values between 0 and 1. Although this is linked to the original work by Jarvis & McNaughton (i.e. the decoupling factor), it seems rather confusing for readers of this manuscript and I would suggest using a consistent terminology throughout the manuscript, e.g. the degree of coupling with high omega values referring to a lower degree of coupling.

A: We are happy to switch our use of terminology to "degree of coupling", noting that these terms are used interchangeably widely across the literature.

R: (2) As the manuscript heavily relies on turbulence based measurements from FLUXNET, there is a high chance that the coupling terminology might be misunderstood. It would help and strengthen the manuscript to more clearly differentiate in the Introduction section, if your terminology of coupling is referring to turbulence conditions above the plant canopy (e.g. quantified by u* or sigma w) or to plant physiological coupling at the leaf level or within the canopy, or between different layers of the canopy such as in forests and woody shrublands. This seems also important to differentiate between the leaf and ecosystem scale in this manuscript as EC flux measurements are at the ecosystem scale, yet some of the presented concepts here are referring to the stomatal coupling at the leaf scale (typically measured by leaf chamber).

A: We do not fully follow the reviewer's lack of clarity on this issue. We define our use

clearly in equation 1, 2 and in particular 3 (which outlines the use of u*). Our approach following Jarvis & others and takes a big-leaf approach. We clearly address potential issues in this approach in our Caveats section (2.3.1). The ecosystem scale is an integration of the leaf-level processes and thus, reference to leaf / canopy processes is appropriate.

R: (3) The manuscript currently relies substantially on comparisons of FLUXNET derived values to the literature, yet the literature values are not presented and analysed quantitatively. I would suggest considering a figure or table comparing both by PFT and documenting details of the so heavily referred to values from the literature, e.g. on how these were assessed/derived (single site/plant experiment, multiple sites, chambers, EC, season etc) to give readers a better idea of their origin and meaning. The manuscript draws substantial conclusions from the comparison to the literature values and these needs to be justified accordingly in a quantitative way that is clearly visualized.

A: We are happy to add such a table that summarises information from the literature we reviewed. We have already indicated we would add this information (i.e. a PFT summary) to Fig 1 in response to reviewer 1.

R: (4) The FLUXNET La Thuile data used here is relatively outdated (from 2007) and only includes a limited number of sites (as Free and Fair use subset). Yet the newer and more extensive FLUXNET2015 dataset is available since late 2015 (same website as referred to in Methods section), but including many more sites and site years compared to the 2007 La Thuile dataset (âĹij1000 vs. âĹij1500 site years), and also including a subset with a similar data policy (TIER1). I am wondering what the reasoning behind this choice of older dataset was and if the manuscript would not benefit from the larger sampling available in the newer dataset, particularly in terms of important PFTs (e.g. TRF) that were poorly represented in the 2007 dataset? It would also benefit the manuscript to have a table of the eventually retained sites (after data screening – see Section 2.1), their used site years and PFT etc. in the Appendix, something that is

typically recommended when using the FLUXNET dataset.

A: We will follow the reviewer's request and add a list of the sites used in the analysis following screening to the appendix.

The FLUXNET2015 release is being made progressively, and hence the data available continue to change on a regular basis. When we originally carried out our analysis, the quality assurance flags for latent heat flux were missing, meaning that we could not carry out our analysis on the new release (a patch has now been released). Owing to the fact that this dataset is still changing, and its properties have not been explored or tested yet, we felt that it was more appropriate at this time to work with the well-known and studied La Thuile dataset. We note that just because there is a newer release, it does not invalidate the approach taken here. We are not the only authors to continue to use the La Thuile data (see for example in Biogeosciences discussions: Mahecha et al. 2017, doi:10.5194/bg-2017-130; Marcolla et al. 2017, doi:10.5194/bg-2017-11).

We have run a similar analysis with the FLUXNET2015 dataset (see figure below). Our conclusions are similar across the two datasets. In particular, the reviewer highlighted the greater number of tropical sites, but as can be seen from our figure, the change in site years is small (n=16 vs. n=9). We will add this as a supplementary figure to demonstrate this.

R: (5) The manuscript correctly states (Section 2.3.1) that soil evaporation would bias the coupling estimates, yet it is assumed that this only matters 24 hours after rainfall. In fact soil evaporation is a substantial component of the measured ET at almost all sites and except in closest canopy forests with high LAI, easily contributes up to 50% of total ET, particularly in grasslands and shrublands. Consequently, the bias of soil evaporation on the results of certain PFTs is likely much higher and this needs to be addressed in the interpretation of the Results.

A: In fact, we screened data 48 hours after rainfall, not 24. There is a discrepancy in our text where we mistakenly state 24 hours in the Caveats section, but 48 in the

method; we will fix this error in the revised version. Of course, our choice of 48 hours is an assumption of the method, but as we highlighted in the Caveats section, it is one that has been widely used (see Law et al., 2002; Groenendijk et al., 2011; Dekker et al., 2016).

As suggested, we will extend our Caveats section to highlight the reviewer's point that this assumption may vary with PFT. However, it is not clear to us where the reviewer's soil evaporation figure of "easily up to 50%" originates; the literature we have read points to transpiration accounting for between 60-80% of evapotranspiration across the land surface (e.g. Miralles et al., 2011; Jasechko et al., 2013; Schlesinger and Jasechko, 2014, but see Schlaepfer et al., 2014).

R: (6) The analysis on the controls of omega is largely focused on wind and precipitation, yet soil moisture and VPD seem much better and more direct controls of plant water stress affecting stomatal conductance. These data are available for most of the sites in the FLUXNET dataset and I would encourage the authors to consider extending their analysis to these controls, and linking these results to the recent literature on stomatal conductance.

A: The effect of VPD is already accounted for through its use in equation 2. With respect to the reviewer's point about soil moisture, the focus of this manuscript was on boundary layer controls on stomatal conductance. There is already ample literature on drought and soil moisture. However, in revision, we will explore the suggested soil moisture output fields but would not expect these to reflect plant water stress for many deep-rooted forest species (FLUXNET fields refer to "upper layer/lower layer" without stating the explicit depth). The reviewer also raises a question below about general variability and it is likely that looking at these data for this question may be more relevant.

R: Overall, I am aware of the length limitations of Opinion & Perspectives papers, yet a full length manuscript might be more fitting for this study to sufficiently document the

analysis and the Conclusions that could be drawn from it.

A: The main goal of this work was to document the degree of coupling observed at FLUXNET sites and demonstrate how it differs from the literature. We feel that the manuscript submission, even with the addition of the new summary table of literature decoupling, sufficiently addresses this goal in its current form. The additions requested by both reviewers do not appear to warrant a substantial extension in length of the paper.

Specific Comments: R: - Page 1, Line 19: please consider adding short explanation why Gs is reduced with elevated CO2.

A: We will add this.

R: - It would help to add some details in Section 2.1. why the flux data were screened this way and how this affects the interpretation of your Results. It would also be helpful to specify that your analysis is presenting mean decoupling values during the peak growing season somewhere in the Results.

A: We will add an explanation sentence of why these data were screened this way. We will also add the requested statement to the results.

R: Page 4, Line 29: why are open grasslands necessarily sites with low precipitation?

A: The reviewer is correct that the open grasslands are not necessarily sites with low PPT; we will reword this sentence.

R: - Page 4, Line 30: or are grasslands just more couple because of having just 1 canopy layer (compared to typically 2 in forest)?

A: This sentence refers to the fact that grasslands at low precipitation are more coupled than grasslands at high precipitation; it does not compare grasslands with forests. Forests are typically more coupled than grasslands.

R: - Page 5, Line 20: please consider removing "low" for consistency.

A: We will remove.

R: - Page 5, Line 21: SDGVM = Sheffield Dynamic Global Vegetation Model (add Global)

A: We will add the missing global.

R: - Page 5, Line 30: it seems incorrect to write "all" FLUXNET sites her, as you are (i) only using a subset from the 2007 dataset and (ii) further reduce this subset by data screening (see Section 2.1).

A: We will replace "all" with "175 sites and 634 site-years".

R: - Page 5, Line 30: I would argue that "forest species" is not the correct term here as you are referring to PFTs, not species groups, and the flux measurements are at the ecosystem scale.

A: We will replace "species" with "PFTs".

R: - Page 5, Line 31: consider limiting "..the FLUXNET network.." to "FLUXNET".

A: We will remove "network".

R: - Page 6, Line 26-27: Ref. Knauer et al. missing in Reference list, and similarly the incomplete citation of Knauer et al. in Line 31-32.

A: We will fix the Knauer et al. reference

R: - Page 6, Line 32: "that" seems redundant here

A: We will remove "that".

R: Section 2.3.1: what about the limitations arising from the use of an older dataset (despite availability of newer dataset, which poorly represents some PFTs?

A: See earlier response about FLUXNET 2015.

R: Page 7, Line 8-9: what about general variability of environmental conditions and

water availability?

A: We agree with the reviewer that anything that alters Gs and thus the ratio of Gs to Ga, will also affect coupling. As stated above, in revision we will attempt to look at whether the FLUXNET outputs for soil moisture provide any additional constraint upon this.

R: Page 7, Line 11: the BADM data of the new FLUXNET dataset is more extensive then previously and includes details on canopy height and LAI for many sites

A: These data are not sufficient to probe the questions we posed, in many cases, particular with canopy/tower height, this information is simply not available at all sites (presumably this is covered by: (i) "At present only the variables of Site_General_Info and Disturbance_and_Management are made available; and (ii) "Additional BADM variables such as LAI, biomass measurements and soil characteristics will be added to the BADM files over time"). The LAI information is also problematic: we do not know how or when these data were measured (LAI-2000, hemispherical photography, other?), we do not know if they are LAI or really plant area index (i.e. not corrected for a woody component, or clumping), we do not know the sampling footprint these data represent and finally we cannot trace the origins of these data. For these reasons, we chose to instead analyse the decoupling in relation to precipitation (a proxy for LAI). We included a figure in response to reviewer 1 to demonstrate some agreement with our figure 3. However, due to the issues we raise above we feel it is more appropriate to stick with our analysis framework. If the reviewer wishes we could include this in the supplementary.

R: Page 7, Line 16-17: please specify how process understanding from leaf to canopy scale can be improved, if all the listed measurements are referring to the individual plant and ecosystem scale. Furthermore, such targeted Gs measurements have been performed at various sites already and it is not clear to me what new aspects the authors are suggesting here.

A: We will add further text to develop the point about leaf to canopy scaling and how additional measurements could be used to improve our understanding. In terms of gs data at various sites: it is true that there is some (8 sites - Medlyn et al. 2017, New Phytologist). However, these data are few and not freely available as part of FLUXNET. Moreover, the temporal and spatial footprint is often not sufficient to address the question we are raising, in no small part because these data were collected with different research questions in mind.

R: Figure 1: C4 PFTs in caption but not displayed in Figure? Please add missing data or specify why these are not displayed. Ditto in Figure A1.

A: We will remove C4 from the caption; this was a mistake, as FLUXNET data does not distinguish between C3 and C4 pathways.

R: - Figure 2: please consider (i) moving site names outside graph as axis caption (i.e. this is a categorical axis), (ii) separating the three groups a-c by vertical lines, (iii) removing selective ticks on x-axis OR adding one for every single site, and (iv) adding details on the meaning of the whiskers in the caption text.

A: We will change the figure as suggested and add the missing caption text.

R: - Figure 3: please consider changing the colours so that these are easier to differentiate, and to change the symbols (i.e. different symbol for each PFT, and potentially increasing size). It could also help to differentiate each regression line with dashed/dotted display.

A: We will explore the reviewer's suggestion when revising this figure.

R: Figure 4: why are the C3 grasses displayed in Fig. 3, yet not here? Also, what about croplands? I would also suggest to consider add the slope values here and in Fig. 3 for the regression lines.

A: The aim of figure 4 was to probe the relationship between wind speed and coupling for forest PFTs. We will make this distinction clearer in revision.

[Figure]

[Figure]

**Fig. 1.** As in figure 1 but using FLUXNET2015 data.

---

## Author Response (AR1)

A summary of the key changes are as follows:

-   Added literature summary table from 40 studies covering 57 PFTs.
-   Added site summary table to the appendix (A2).
-   Adjusted figure symbols, legend, layout in Figs. 2 & 3 as requested. We have also coloured forest PFT types in Fig. 4 to assist the reader.
-   Added missing C4C category to fig. 1, A1 & A2.
-   As we do not attempt to define C4 grass fractions at flux sites, we rename the grass PFT (GRA), where previously it had been labelled C3 Grass (C3G).
-   Added additional sites that were mistakenly screened to Fig. 3. This does not alter the original patterns or conclusions drawn (see below).
-   Corrected the number of sites and site-years reported in the methods. In the original submission these were calculated before we excluded mixed forest, wetland and unclassified PFTs and so did not match the total shown in Fig. 1.

1    We again thank the reviewer for their constructive comments and we address their

2    various concerns below. This version replaces our original response, updated with

3    specific changes where appropriate. Referee comments are highlighted in red, with

4    our response below in black in each case.

5    R: This paper leverages the new FLUXNET2015 dataset to estimate differences in the

6    decoupling coefficient across plant funcitonal types, with some additional discussion

7    of how the coefficient varies in response to canopy structure and meteorological

8    condition. The work builds off a previous study that highlighted the decoupling

9    coefficient as a significant source of uncertainty in some model predictions (De

10   Kauwe et al. 2013). The authors report that evergreen forests are more decoupled than

11   previously thought, and that the decoupling of grasslands depends on mean annual

12   precipitation (among other results).

14   Overall, I think this analysis will be of interest to members of the observational and

15   modeling communities, and the article is generally well written and the figures are

16   clearly presented. I do have a few suggestions for the authors that would allow them

17   to bridge what I perceive to be a bit of a gap between the rational/objective of the

18   paper and the interpretation of results.

20   First, the authors aimed to "examine if decoupling coefficients from FLUXNET were

21   consistent with the literature values." However, the comparison of the decoupling

22   coefficients derived from FLUXNET data and literature values was largely

23   qualitative. The comparison would be more informative if values reported in the

24   literature (or assumed by the models) were presented alongside those derived from the

25   Flux data (for example, by including a bit more information in the box and whisker

26   plots of Figure 1).

27   A: To address the reviewer's point about our comparison being qualitative, we have

28   now added a comprehensive literature review. We have summarised estimates of the

29   decoupling coefficient from across 40 studies, covering 57 PFTs. These results can be

30   seen in our new Table 1. On reflection however, we do not feel it is appropriate to add

31   this information to Fig 1. We feel that adding additional bars would give a false

equivalence, owing to the significantly smaller number of sites obtained from the literature review.

We agree with the reviewer that information on what is currently assumed by models would be a nice addition to the figure; however, this information is not available as it is rarely (if ever) reported. We recommended this information be reported following our analysis of several models in a recent model intercomparison (De Kauwe et al. 2013, Global Change Biology) but our insight into model assumptions is still limited to the models considered there.

R: Second, the authors aimed to "develop a benchmark metric against which to test model assumptions about decoupling." Presumably this "benchmark metric" is the range of decoupling coefficients presented in the results. Would it be possible for the authors to demonstrate, at least at a few sites, that using a decoupling coefficient informed by the results of this study indeed improves agreement between the predictions of at least one model, and observations (for example, flux tower observations of ET)?

A: Whilst we obviously see value in what the reviewer has suggested, we feel this is actually a very separate piece of research. What we have aimed to do here was to raise the issue again in the literature by developing a potential benchmarking metric (Fig. 1) for models. We have written this up as an "ideas and perspective" and not a full research paper specifically for this reason. The next logical step would be to determine what state-of-the-art models are currently assuming by comparison, but that needs to be an exhaustive study. Little insight would be gained by playing with one model quickly in isolation: models often get the right answers for the wrong reasons, and we would want to guard against this.

To the route forward section, we have added: "The next steps involve determining what models currently assume about the degree of coupling and then to determine how flux-derived estimates of coupling would change model predictions."

 R: I was also curious about the author's choice to limit the analysis to relatively windy periods between 800 and 1600 hours. Coupling should be greater during these condition when compared to relatively stable conditions, for example those experienced from late evening to sunrise. Do the models similarly use a decoupling coefficient that is most appropriate for those conditions, or do they perhaps employ a lower value that is representative of daytime and nighttime periods (particularly if the models run at a daily timestep)?

A: This is an interesting question. The simple answer is that we deemed daylight hours and timesteps with $u_* < 0.25$ to be the period when stomatal control over transpiration would be strongest. Additionally, we would assume that in more stable conditions the errors in FLUXNET data would be greater (due to reduced turbulence), so have avoided this period.

R: Further, I though the authors might have missed an opportunity to leverage the high-frequency data from FLUXNET to say something about temporal variation in decoupling over the course of a typical day.

A: This also would be a valuable direction for further research. Decoupling factors are not constant: both models and data will show variation during the course of the day. Our goal here was simply to examine how long-term average decoupling factors compare with literature values. Future more detailed work could examine the sub-diurnal variation in both models and measurements.

We have added to the Route forward section: "In this study we examined the long-term average coupling factor. It may also be instructive to consider how estimated coupling factors change across the course of a day or within a season. However, it is likely that such an approach may be more sensitive to noise in the fluxes as well as events such as drought."

R: Finally, in paragraph 10, the authors state that LAI information for many sites is not available. Many FLUXNET sites have high-quality ground-based LAI

93 measurements that are not reported to the network. Sometimes an email to the site PIs

94 can turn up useful ancillary data.

95 A: Firstly we note that the recommendation to email site PI's is unrealistic given our

96 analysis covered 164 sites. We did carry out the suggested analysis using an ancillary

97 file of LAI data from the FLUXNET sites. We have a number of concerns about

98 presenting these data however. The file we used is no longer available online, and we

99 have little information about what the data represent. We do not know how or when

100 these data were measured (LAI-2000, hemispherical photography, other?), we do not

101 know if they are LAI or really plant area index (i.e. not corrected for a woody

102 component, or clumping), we do not know the sampling footprint these data represent

103 and finally we cannot trace the origins of these data. For these reasons, we chose to

104 instead analyse the decoupling in relation to precipitation (a proxy for LAI). We show

105 the reviewer this figure below and as they can see there is some agreement with our

106 figure 3, but due to the issues we raise above we feel it is more appropriate to stick

107 with our analysis framework. If the reviewer wishes we could include this in the

108 supplementary.

109

110

111

112

We again thank the reviewer for their constructive comments and we address their various concerns below. This version replaces our original response, updated with specific changes where appropriate. Referee comments are highlighted in red, with our response below in black in each case.

R: This manuscript presents results from a FLUXNET based analysis on vegetation-atmosphere coupling of transpiration using the omega factor by Jarvis & McNaughton. Aggregating daytime data during the peak growing season across plant functional types (PFT), it was found that evergreen needleleaf forests (ENF) have a lower degree of coupling, and that evergreen broadleaf forests (EBF) and shrubs were more coupled then previously suggested in the literature. The manuscript concludes that this decoupling analysis based on FLUXNET data can be used for benchmarking to test models. The manuscript is overall well written (particularly the Discussion section) and the presented research is of significant scientific interest to improve model estimates of biosphere-atmosphere exchange. Nonetheless, I do have some concerns regarding the argumentation and analysis presented here and would strongly encourage the authors to consider the following points, before a revised manuscript could be recommended for publication.

Main Points:

R: (1) While the manuscript is overall focused on the coupling of vegetation and atmosphere regarding transpiration, the manuscript incoherently switches between the use of the degree of coupling and decoupling, which all refer to omega values between 0 and 1. Although this is linked to the original work by Jarvis & McNaughton (i.e. the decoupling factor), it seems rather confusing for readers of this manuscript and I would suggest using a consistent terminology throughout the manuscript, e.g. the degree of coupling with high omega values referring to a lower degree of coupling.

A: We are happy to switch our use of terminology to "degree of coupling", noting that these terms are used interchangeably widely across the literature.

R: (2) As the manuscript heavily relies on turbulence based measurements from FLUXNET, there is a high chance that the coupling terminology might be misunderstood. It would help and strengthen the manuscript to more clearly differentiate in the Introduction section, if your terminology of coupling is referring to turbulence conditions above the plant canopy (e.g. quantified by u* or sigma w) or to plant physiological coupling at the leaf level or within the canopy, or between different layers of the canopy such as in forests and woody shrublands. This seems also important to differentiate between the leaf and ecosystem scale in this manuscript as EC flux measurements are at the ecosystem scale, yet some of the presented concepts here are referring to the stomatal coupling at the leaf scale (typically measured by leaf chamber).

A: We do not fully follow the reviewer's lack of clarity on this issue. We define our use clearly in equation 1, 2 and in particular 3 (which outlines the use of u*). Our approach following Jarvis & others and takes a big-leaf approach. We clearly address potential issues in this approach in our Caveats section (2.3.1). The ecosystem scale is an integration of the leaf-level processes and thus, reference to leaf / canopy processes is appropriate.

R: (3) The manuscript currently relies substantially on comparisons of FLUXNET derived values to the literature, yet the literature values are not presented and analysed quantitatively. I would suggest considering a figure or table comparing both by PFT and documenting details of the so heavily referred to values from the literature, e.g. on how these were assessed/derived (single site/plant experiment, multiple sites, cham- bers, EC, season etc) to give readers a better idea of their origin and meaning. The manuscript draws substantial conclusions from the comparison to the literature values and these needs to be justified accordingly in a quantitative way that is clearly visualized.

A: We have now summarised the literature from 40 studies, covering 57 PFTs, in our new table 1.

R: (4) The FLUXNET La Thuile data used here is relatively outdated (from 2007) and only includes a limited number of sites (as Free and Fair use subset). Yet the newer

and more extensive FLUXNET2015 dataset is available since late 2015 (same website as referred to in Methods section), but including many more sites and site years compared to the 2007 La Thuile dataset (~1000 vs. ~1500 site years), and also including a subset with a similar data policy (TIER1). I am wondering what the reasoning behind this choice of older dataset was and if the manuscript would not benefit from the larger sampling available in the newer dataset, particularly in terms of important PFTs (e.g. TRF) that were poorly represented in the 2007 dataset? It would also benefit the manuscript to have a table of the eventually retained sites (after data screening – see Section 2.1), their used site years and PFT etc. in the Appendix, something that is typically recommended when using the FLUXNET dataset.

A: We have added the list of the sites used in the analysis following screening to the appendix.

The FLUXNET2015 release is being made progressively, and hence the data available continue to change on a regular basis. When we originally carried out our analysis, the quality assurance flags for latent heat flux were missing, meaning that we could not carry out our analysis on the new release (a patch has now been released). Owing to the fact that this dataset is still changing, and its properties have not been explored or tested yet, we felt that it was more appropriate at this time to work with the well-known and studied La Thuile dataset. We note that just because there is a newer release, it does not invalidate the approach taken here. We are not the only authors to continue to use the La Thuile data (see for example in Biogeosciences discussions: Mahecha et al. 2017, doi:10.5194/bg-2017-130; Marcolla et al. 2017, doi:10.5194/bg-2017-11).

We have run a similar analysis with the FLUXNET2015 dataset (see our new Figure A1). Our conclusions are similar across the two datasets. In particular, the reviewer highlighted the greater number of tropical sites, but as can be seen from our figure, the change in site years is small (n=16 vs. n=9).

To the end of the methods section we have added: "We also replicated our analysis using eddy covariance data taken from the FLUXNET2015 dataset (http://fluxnet.fluxdata.org/data/fluxnet2015-dataset). Figure A1 is a replicate of Fig. 1 and shows the patterns we derived are robust across flux releases."

R: (5) The manuscript correctly states (Section 2.3.1) that soil evaporation would bias the coupling estimates, yet it is assumed that this only matters 24 hours after rainfall. In fact soil evaporation is a substantial component of the measured ET at almost all sites and except in closest canopy forests with high LAI, easily contributes up to 50% of total ET, particularly in grasslands and shrublands. Consequently, the bias of soil evaporation on the results of certain PFTs is likely much higher and this needs to be addressed in the interpretation of the Results.

A: In fact, we screened data 48 hours after rainfall, not 24. There is a discrepancy in our text where we mistakenly state 24 hours in the Caveats section, but 48 in the method; we have fixed this error in the revised version. Of course, our choice of 48 hours is an assumption of the method, but as we highlighted in the Caveats section, it is one that has been widely used (see Law et al., 2002; Groenendijk et al., 2011; Dekker et al., 2016).

As suggested, we now highlight in our Caveats section to highlight the reviewer's point that this assumption may vary with PFT. However, it is not clear to us where the reviewer's soil evaporation figure of "easily up to 50%" originates; the literature we have read points to transpiration accounting for between 60-80% of evapotranspiration across the land surface (e.g. Miralles et al., 2011; Jasechko et al., 2013; Schlesinger and Jasechko, 2014, but see Schlaepfer et al., 2014).

R: (6) The analysis on the controls of omega is largely focused on wind and precipitation, yet soil moisture and VPD seem much better and more direct controls of plant water stress affecting stomatal conductance. These data are available for most of the sites in the FLUXNET dataset and I would encourage the authors to consider extending their analysis to these controls, and linking these results to the recent literature on stomatal conductance.

A: The effect of VPD is already accounted for through its use in equation 2. With respect to the reviewer's point about soil moisture, the focus of this manuscript was on boundary layer controls on stomatal conductance. There is already ample literature on drought and soil moisture.

Nevertheless, as requested by the reviewer, we did explore the soil moisture fields:

(1)   We were unable to determine what depths "upper layer" and "lower layer" refer to FLUXNET, or if these are consistent across sites, we presume not?

(2)   Setting an arbitrary threshold of acceptable data to be at least 20% of a given year, we found that there were only 11 sites with data from the upper layer and 9 sites with data from the lower layer. These site numbers are reduced further to 7 and 5 for the lower and upper layers, respectively, if one assumes sites: CA-NS1, CA-NS2, CA-NS3, CA-NS4, CA-NS6 show (presumably close to) the same information.

Given these stated reasons, we have not pursued any further analysis related to soil moisture.

R: Overall, I am aware of the length limitations of Opinion & Perspectives papers, yet a full length manuscript might be more fitting for this study to sufficiently document the analysis and the Conclusions that could be drawn from it.

A: The main goal of this work was to document the degree of coupling observed at FLUXNET sites and demonstrate how it differs from the literature. We feel that the manuscript submission, even with the addition of the new summary table of literature decoupling, sufficiently addresses this goal in its current form. The additions requested by both reviewers do not appear to warrant a substantial extension in length of the paper.

Specific Comments:

R: - Page 1, Line 19: please consider adding short explanation why Gs is reduced with elevated CO2.

A: We have added: "due to either a decrease in stomatal aperture with the reduced photosynthetic demand for $CO_2$ and/or a change in stomatal density (McElwain and Chaloner, 1995; Woodward and Kelly, 1995)".

R: - It would help to add some details in Section 2.1. why the flux data were screened this way and how this affects the interpretation of your Results. It would also be helpful to specify that your analysis is presenting mean decoupling values during the peak growing season somewhere in the Results.

A: For each of the screening choices we have now added an explanation: "Flux data were first screened as follows: (i) data flagged as "good" (quality control flag "fqcOK" = 1; Williams et al., 2012); (ii) data from the three most productive months, to account for the different timing of summer in the Northern and Southern hemispheres; (iii) daylight hours between 8 am and 4 pm, to account for periods when the vegetation is photosynthesising; (iv) half-hours with precipitation, and the subsequent 48 half-hours, were excluded to minimise the influence of soil evaporation (Law et al., 2002; Groenendijk et al., 2011; Dekker et al., 2016); and (v) data with a u∗ < 0.25 were excluded to avoid conditions of low turbulence (Sánchez et al., 2010)."

We have also added to the methods: "In our analysis we derived the average (three most productive months) decoupling coefficient, as the focus of our manuscript was on the spatial variability in coupling across FLUXNET. This is likely to be a metric that can most readily be exploited to assess existing coupling assumptions in models. Future analysis may wish to explore the temporal variability in this metric."

R: Page 4, Line 29: why are open grasslands necessarily sites with low precipitation?

A: The reviewer is correct that the open grasslands are not necessarily sites with low PPT. We have reworded this sentence to be: "The data suggest that for sites that are likely to be more open grasslands (i.e. sites with a low precipitation)…"

187

R: - Page 4, Line 30: or are grasslands just more couple because of having just 1 canopy layer (compared to typically 2 in forest)?

A: This sentence refers to the fact that grasslands at low precipitation are more coupled than grasslands at high precipitation; it does not compare grasslands with forests. Forests are typically more coupled than grasslands.

R: - Page 5, Line 20: please consider removing "low" for consistency.

A: We have removed "low".

R: - Page 5, Line 21: SDGVM = Sheffield Dynamic Global Vegetation Model (add Global)

A: We have added the missing "global".

R: - Page 5, Line 30: it seems incorrect to write "all" FLUXNET sites her, as you are (i) only using a subset from the 2007 dataset and (ii) further reduce this subset by data screening (see Section 2.1).

A: We have replaced "all" with "the 164 FLUXNET sites".

R: - Page 5, Line 30: I would argue that "forest species" is not the correct term here as you are referring to PFTs, not species groups, and the flux measurements are at the ecosystem scale.

A: We have replaced "species" with "PFTs".

R: - Page 5, Line 31: consider limiting "..the FLUXNET network.." to "FLUXNET".

A: We have removed "network".

 R: - Page 6, Line 26-27: Ref. Knauer et al. missing in Reference list, and similarly the
215 incomplete citation of Knauer et al. in Line 31-32.

216 A: We have fixed the Knauer et al. reference

217

218 R: - Page 6, Line 32: "that" seems redundant here

219 A: We have removed "that".

220

221 R: Section 2.3.1: what about the limitations arising from the use of an older dataset
222 (despite availability of newer dataset, which poorly represents some PFTs?

223 A:  See earlier response about FLUXNET 2015.

224

225 R:  Page 7, Line 8-9: what about general variability of environmental conditions and
226 water availability?

227 A: We agree with the reviewer that anything that alters Gs and thus the ratio of Gs to
228 Ga, will also affect coupling. As previously stated, the focus of our analysis was on

229 boundary layer controls on stomatal conductance. We were interested in determining
230 if we could extract a metric related to coupling with which existing model
231 assumptions could be probed. As our interest was related to variability in space, we
232 feel our approach was the correct first step.

233

234 R: Page 7, Line 11: the BADM data of the new FLUXNET dataset is more extensive
235 then previously and includes details on canopy height and LAI for many sites

236 A: These data are not sufficient to probe the questions we posed, in many cases,
237 particular with canopy/tower height, this information is simply not available at all
238 sites (presumably this is covered by: (i) "At present only the variables of
239 Site_General_Info and Disturbance_and_Management are made available; and (ii)
240 "Additional BADM variables such as LAI, biomass measurements and soil
241 characteristics will be added to the BADM files over time"). The LAI information is
242 also problematic: we do not know how or when these data were measured (LAI-2000,

hemispherical photography, other?), we do not know if they are LAI or really plant

area index (i.e. not corrected for a woody component, or clumping), we do not know

the sampling footprint these data represent and finally we cannot trace the origins of

these data. For these reasons, we chose to instead analyse the decoupling in relation to

precipitation (a proxy for LAI). We included a figure in response to reviewer 1 to

demonstrate some agreement with our figure 3. However, due to the issues we raise

above we feel it is more appropriate to stick with our analysis framework. If the

reviewer wishes we could include this in the supplementary.

R: Page 7, Line 16-17: please specify how process understanding from leaf to canopy

scale can be improved, if all the listed measurements are referring to the individual

plant and ecosystem scale. Furthermore, such targeted Gs measurements have been

performed at various sites already and it is not clear to me what new aspects the

authors are suggesting here.

A: We have now added: "Recently, Medlyn et al. (2017) compared estimates of plant

water-use efficiency derived from leaf gas exchange data and eddy flux data for eight

sites where these measurements were acquired at the same point in time. They found

similarities for DBF and TRF PFTs, but differences for EBF and ENF PFTs. The

authors were unable to explain these scaling discrepancies. Further targeted

measurements campaigns at flux sites could lead to new knowledge, which would

advance our understanding of the processes involved in scaling from the leaf to the

canopy."

R: Figure 1: C4 PFTs in caption but not displayed in Figure? Please add missing data

or specify why these are not displayed. Ditto in Figure A1.

A: We have added the missing C4C category that was mistakenly not previously

shown. For grasses, as we do not separate the C4 fraction (FLUXNET does not

provide enough information), we now label all grasses as GRA (i.e. not C3G).

R: - Figure 2: please consider (i) moving site names outside graph as axis caption (i.e.

this is a categorical axis), (ii) separating the three groups a-c by vertical lines, (iii)

274 removing selective ticks on x-axis OR adding one for every single site, and (iv)
275 adding details on the meaning of the whiskers in the caption text.

276 A: We have changed the figure as suggested and add the missing caption text.

277

278 R: - Figure 3: please consider changing the colours so that these are easier to differ-
279 entiate, and to change the symbols (i.e. different symbol for each PFT, and poten-
280 tially increasing size). It could also help to differentiate each regression line with
281 dashed/dotted display.

282 A: In revising the figure we realised that we had accidently screened out some site
283 based on our calculation of precipitation in the most productive months. The new
284 relationships shown in Figure 3 between the decoupling coefficient and precipitation
285 are consistent with the original submission, although perhaps unexpectedly, including
286 more sites increases the variability in the relationship. We have amended the text
287 accordingly.

288

289 As requested we have attempted to make the figure easier to interpret: we have
290 removed the transparency to make the symbols bolder, we have used different symbol
291 types and simplified the legend.

292

293 R: Figure 4: why are the C3 grasses displayed in Fig. 3, yet not here? Also, what
294 about croplands? I would also suggest to consider add the slope values here and in
295 Fig. 3 for the regression lines.

296 A: The aim of figure 4 was to probe the relationship between wind speed and
297 coupling for forest PFTs. To make this distinction clearer, we have individually

[revised manuscript text omitted]